# CLOSED-LOOP DATA TRANSCRIPTION TO AN LDR VIA MINIMAXING RATE REDUCTION

## ABSTRACT

This work proposes a new computational framework for automatically learning a closed-loop transcription between multi-class multi-dimensional data and a *linear discriminative representation (LDR)* that consists of multiple multi-dimensional linear subspaces. In particular, we argue that the optimal encoding and decoding mappings sought can be formulated as the equilibrium point of a *two-player minimax game between the encoder and decoder*. A natural utility function for this game is the so-called *rate reduction*, a simple information-theoretic measure for distances between mixtures of subspace-like Gaussians in the feature space. Our formulation avoids expensive evaluating and minimizing approximated distances between arbitrary distributions in either the data space or the feature space. To a large extent, conceptually and computationally this new formulation unifies the benefits of Auto-Encoding and GAN and naturally extends them to the settings of learning a *both discriminative and generative* representation for complex multi-class and multi-dimensional real-world data. Our extensive experiments on many benchmark datasets demonstrate tremendous potential of this framework: under fair comparison, visual quality of the learned decoder and classification performance of the encoder is competitive and often better than existing methods based on GAN, VAE or a combination of both.

## 1 INTRODUCTION AND RELATED WORK

One of the most fundamental tasks in machine learning is to learn and model complex distributions (or structures) of real-world data, such as images or texts, from a set of observed samples. By "learn and model", one typically means that we want to establish a (parameteric) mapping between the distribution of the real data, say $\boldsymbol{x} \in \mathbb{R}^D$, and a more compact random variable, say $\boldsymbol{z} \in \mathbb{R}^d$:

$$f(\cdot, \theta) : \boldsymbol{x} \in \mathbb{R}^D \mapsto \boldsymbol{z} \in \mathbb{R}^d \quad \text{or the inverse} \quad g(\cdot, \eta) : \boldsymbol{z} \in \mathbb{R}^d \mapsto \boldsymbol{x} \in \mathbb{R}^D, \tag{1}$$

where $\boldsymbol{z}$ has certain standard structure or distribution (e.g. normal distributions). The so-learned representation or feature $\boldsymbol{z}$ would be much easier to use for either generative or discriminative purposes. *Be aware* that the support of the distribution of $\boldsymbol{x}$ (and that of $\boldsymbol{z}$) can be low-dimensional hence the above map(s) may not be so well defined off the support in the space $\mathbb{R}^D$ (or $\mathbb{R}^d$).

### 1.1 LEARNING GENERATIVE MODELS VIA AUTO-ENCODING OR GAN

**Auto-Encoding.** In the machine learning literature, roughly speaking, there have been two representative approaches to such a distribution-learning task. One is the classic "Auto Encoding" (AE) approach (Kramer, 1991; Hinton & Zemel, 1993) that aims to simultaneously learn an encoding mapping $f$ from $\boldsymbol{x}$ to $\boldsymbol{z}$ and an (inverse) decoding mapping $g$ from $\boldsymbol{z}$ back to $\boldsymbol{x}$:

$$\boldsymbol{X} \xrightarrow{f(\boldsymbol{x},\theta)} \boldsymbol{Z} \xrightarrow{g(\boldsymbol{z},\eta)} \hat{\boldsymbol{X}}. \tag{2}$$

Here we use bold capital letters to indicate a matrix of finite samples $\boldsymbol{X} = [\boldsymbol{x}^1, \ldots, \boldsymbol{x}^n] \in \mathbb{R}^{D \times n}$ of $\boldsymbol{x}$ and their mapped features $\boldsymbol{Z} = [\boldsymbol{z}^1, \ldots, \boldsymbol{z}^n] \subset \mathbb{R}^{d \times n}$, respectively. Typically, one wishes for two properties: firstly, the (empirical) distribution of the mapped samples $\boldsymbol{Z}$, denoted as $\hat{p}(\boldsymbol{z} \mid \boldsymbol{X}, \theta)$, is close to certain desired distribution $p(\boldsymbol{z})$, say some much lower-dimensional disentangled multivariate Gaussian;[1] and, secondly, the decoded samples $\hat{\boldsymbol{X}}$ are "similar" or close to the original $\boldsymbol{X}$.

In the feature space $\boldsymbol{z}$, to enforce the learned distribution $\hat{p}(\boldsymbol{z})$ to be close to a target $p(\boldsymbol{z})$, one normally minimizes a certain "distance" between the two distributions, say the *KL-divergence*: $\min \mathcal{D}_{KL}(\hat{p}, p)$. However, it is very difficult, often computationally intractable, to precisely compute or minimize such a distance $\mathcal{D}$ between two arbitrary degenerate distributions in high-dimensional

---

[1]The classical PCA can be viewed as a special case of this task. In fact, the original auto-encoding is precisely justified as *nonlinear* PCA (Kramer, 1991).

spaces. So in practice, one typically chooses to minimize instead certain approximate bounds (surrogates) of the distance, such as a variational bound in *variational auto-encoding* (VAE) (Kingma & Welling, 2013; Zhao et al., 2017) and the *earth mover's (Wasserstein) distance*. In this work, we will show that if we impose specific requirements on the (distribution of) learned feature $z$ (e.g. being subspace-like Gaussians), *natural distances can be given in closed-form.*

**GAN.**   Compared to measuring distribution distance in the (often controlled) feature space $z$, a much more challenging issue is how to effectively measure the distance between the decoded samples $\hat{X}$ and the original $X$ in the data space $x$. For instance, for visual data such as images, despite extensive studies in the computer vision and image processing literature (Wang et al., 2004), it remains elusive to find a good measure for similarity of real images that is both efficient to compute and effective in capturing visual quality and semantic information of the images equally well. *Generative Adversarial Nets (GAN)* (Goodfellow et al., 2014) offers an ingenious idea to resolve this difficulty by utilizing a powerful discriminator $d$, usually modeled and learned by a deep network, to discern differences between the generated samples $\hat{X}$ and the real ones $X$:

$$Z \xrightarrow{g(z,\eta)} \hat{X}, X \xrightarrow{d(x,\theta)} Y \in {0, 1}\}. \tag{3}$$

Such a discriminator minimizes the Jensen-Shannon divergence, between the data $X$ and $\hat{X}$. But as shown in (Arjovsky et al., 2017), the JS-divergence can be highly ill-conditioned to optimize when the distributions are low-dim. So in practice, one may choose to replace the JS-divergence with the earth mover's distance or other variants. The original GAN aims to directly learn a mapping $g$, called a generator, from a standard distribution (say a low-dim Gaussian random field) to the real (visual) data distribution in a high-dim space. However, distributions of real-world data can be rather sophisticated and often contain *multiple* classes and *multiple* factors in each class (Bengio et al., 2013). That makes learning the mapping $g$ rather challenging in practice, suffering difficulties such as *mode-collapse* (Srivastava et al., 2017). As a result, many variants of GAN have been subsequently developed in order to improve the stability and performance in learning multiple modes and disentangling different factors in the data distribution, such as *Conditional GAN* (Mirza & Osindero, 2014; Sohn et al., 2015; Mathieu et al., 2016; Van den Oord et al., 2016; Wang et al., 2018), *InfoGAN* (Chen et al., 2016; Tang et al., 2021), or *Implicit Maximum Likelihood Estimation (IMLE)* (Li & Malik, 2018; Li et al., 2020)

In particular, to learn a generator for multi-class data, prevalent conditional GAN literature requires label information as inputs during training (Mirza & Osindero, 2014; Odena et al., 2017; Dumoulin et al., 2016; Brock et al., 2018). Recently Wu et al. (2019b;a) has proposed to train a $k$-class GAN by generalizing the two-class cross entropy to a $(k + 1)$-class cross entropy. In this work, *we will introduce a more refined $2k$-class measure* for the $k$ real and $k$ generated classes. In addition, to avoid features for each class to collapse to a singleton, *we will use the so-called rate reduction measure that promotes multi-dimension in the learned features* (Yu et al., 2020).

Another line of research is about how to stablize the training of GAN. SN-GAN (Miyato et al., 2018) has shown spectral normalization on the discriminator is rather effective, which we will adopt in our work. PacGAN (Lin et al., 2018) shows that the training stability can be significantly improved by packing a pair of real and fake images together for the discriminator. In this work, *we show how to naturally generalize this idea to discriminating an arbitrary number of pairs of real and decoded samples.* Also, Wu et al. (2019a) has shown that optimizing the latent features leads to state of the art visual quality. There are strong reasons to believe that their method essentially utilizes the Compressed Sensing principle to implicitly exploit low-dimensionality of the feature distribution. Our framework *will explicitly impose and exploits such low-dimensional structures of the learned feature distribution.*

**Combination of AE and GAN.**   Although AE (VAE) and GAN have started with somewhat different motivations, they have evolved into popular and effective frameworks for learning and modeling complex distributions of many real-world data such as images. Many recent efforts tend to combine both Auto-Encoding and GAN to generate more powerful generative frameworks for more diverse data sets, such as Larsen et al. (2015); Rosca et al. (2017); Srivastava et al. (2017); Bao et al. (2017); Huang et al. (2018); Ulyanov et al. (2018); Vahdat & Kautz (2020). As we will see, in our framework, AE and GAN can be naturally interpreted as two segments of a closed-loop data transcription process. But unlike GAN or VAE, the distribution of the feature $z$ is learned from the data $x$ and *its low-dim support in $\mathbb{R}^d$ is explicitly modeled as a union of discriminative subspaces.*

## 1.2 Learning Linear Discriminative Representation via Rate Reduction

Recently, Chan et al. (2021) has proposed a new objective for deep learning that aims to learn a *linear discriminative representation* (LDR) for multi-class data. The basic idea is to map distributions of real data, potentially on *multiple* nonlinear submanifolds, to a family of canonical models consisting of multiple incoherent (or orthogonal) linear subspaces. To some extent, this generalizes nonlinear PCA (Kramer, 1991) to the more general/realistic settings where we simultaneously apply *multiple nonlinear PCAs* to data on multiple nonlinear submanifolds. Unlike conventional discriminative methods that only aim to predict class labels, the LDR aims to learn the likely multi-dimensional distribution of the data hence is potentially suitable for both discriminative and generative purposes. It has been shown that this can be achieved via maximizing the so-called "rate reduction" objective.

**MCR$^2$.** More precisely, consider a set of data samples $\boldsymbol{X} = [\boldsymbol{x}^1, \ldots, \boldsymbol{x}^n] \in \mathbb{R}^{D \times n}$ from $k$ different classes and we use $\boldsymbol{\Pi}^j, j = 1, \ldots, k$ to denote the memberships of the samples in $k$ classes $\boldsymbol{X} = \cup_{j=1}^k \boldsymbol{X}_j$. One seeks a continuous mapping $f(\cdot, \theta) : \boldsymbol{x} \mapsto \boldsymbol{z}$ from $\boldsymbol{X}$ to an optimal representation $\boldsymbol{Z} = [\boldsymbol{z}^1, \ldots, \boldsymbol{z}^n] \subset \mathbb{R}^{d \times n}$ that maximizes the following coding rate reduction objective, known as the MCR$^2$ principle (Yu et al., 2020):

$$\max_{\boldsymbol{Z}} \ \Delta R(\boldsymbol{Z}, \boldsymbol{\Pi}, \epsilon) \doteq \underbrace{\frac{1}{2} \log \det \left( \boldsymbol{I} + \alpha \boldsymbol{Z}\boldsymbol{Z}^* \right)}_{R(\boldsymbol{Z}, \epsilon)} - \underbrace{\sum_{j=1}^k \frac{\gamma_j}{2} \log \det \left( \boldsymbol{I} + \alpha_j \boldsymbol{Z}\boldsymbol{\Pi}^j \boldsymbol{Z}^* \right)}_{R_c(\boldsymbol{Z}, \epsilon | \boldsymbol{\Pi})}, \quad (4)$$

where $\alpha = \frac{d}{n\epsilon^2}$, $\alpha_j = \frac{d}{\mathrm{tr}(\boldsymbol{\Pi}^j)\epsilon^2}$, $\gamma_j = \frac{\mathrm{tr}(\boldsymbol{\Pi}^j)}{n}$ for $j = 1, \ldots, k$. In this paper, for simplicity we denote $\Delta R(\boldsymbol{Z}, \boldsymbol{\Pi}, \epsilon)$ as $\Delta R(\boldsymbol{Z})$ assuming $\boldsymbol{\Pi}, \epsilon$ are known and fixed. The first term $R(\boldsymbol{Z} | \epsilon)$ is the coding rate of the whole feature set $\boldsymbol{Z}$ (coded as a Gaussian source) with a prescribed precision $\epsilon$; the second term $R_c(\boldsymbol{Z} | \boldsymbol{\Pi}, \epsilon)$ is the average coding rate of the $k$ subsets of features $\boldsymbol{Z}_j = f(\boldsymbol{X}_j)$ (each coded as a Gaussian). As it has been shown by Yu et al. (2020), maximizing the difference between the two terms will "expand" the whole feature set while "compressing and linearizing" features of each of the $k$ classes. If the mapping $f$ maximizes the rate reduction, it maps the features of different classes into independent (orthogonal) subspaces in $\mathbb{R}^d$. Figure 1 illustrates a simple example of data with $k = 2$ classes (on two submanifolds) mapped to two incoherent subspaces (solid black lines).

## 2 Data Transcription via Rate Reduction

### 2.1 LDR Transcription

One issue with this one-sided LDR learning is that maximizing the above objective tends to maximize the dimension of the learned subspace for features in each class.[2] To verify whether the learned features are good, we may consider learning a decoder $g(\cdot, \eta) : \boldsymbol{z} \mapsto \boldsymbol{x}$ from the representation $\boldsymbol{Z} = f(\boldsymbol{X}, \theta)$ back to the data space $\boldsymbol{x}$: $\hat{\boldsymbol{X}} = g(\boldsymbol{Z}, \eta)$, and check how close $\boldsymbol{X}$ and $\hat{\boldsymbol{X}}$ are or how close their features $\boldsymbol{Z}$ and $\hat{\boldsymbol{Z}}$ are. The overall pipeline can be illustrated by the following diagram:

$$\boldsymbol{X} \xrightarrow{f(\boldsymbol{x}, \theta)} \boldsymbol{Z} \xrightarrow{g(\boldsymbol{z}, \eta)} \hat{\boldsymbol{X}} \xrightarrow{f(\boldsymbol{x}, \theta)} \hat{\boldsymbol{Z}}, \quad (5)$$

where the overall model has parameters: $\Theta = \{\theta, \eta\}$. Notice that in the above process, the segment from $\boldsymbol{X}$ to $(\hat{\boldsymbol{X}}, \boldsymbol{X})$ resembles a typical *Auto-Encoding* process although, as we will soon see, our MCR$^2$-based encoder $f$ plays an additional role as a discriminator. The segment from $\boldsymbol{Z}$ to $(\hat{\boldsymbol{Z}}, \boldsymbol{Z})$ draws resemblance to the typical GAN process although, in our context, the distribution of the latent variable $\boldsymbol{z}$ will be learned from the data $\boldsymbol{x}$, which often is a distribution with multiple modes and each mode with multiple factors.

Here, in the specific context of rate reduction, we name this special auto-encoding process *LDR Transcription* since the maximal rate reduction principle explicitly transcribes the data $\boldsymbol{X}$, via $f$, to features $\boldsymbol{Z}$ on a linear discriminative representation (LDR), which can be subsequently decoded back to the data space $\hat{\boldsymbol{X}}$, via $g$. Hence, the encoding and decoding maps $f$ and $g$ together form a closed-loop process, as illustrated in Figure 1. We wish this closed-loop transcription process to have the following good properties:

- **Injectivity:** the generated $\hat{\boldsymbol{x}} = g(f(\boldsymbol{x}, \theta), \eta) \in \hat{\boldsymbol{X}}$ to be as close to (ideally the same as) the original data $\boldsymbol{x} \in \boldsymbol{X}$, in terms of certain measure of similarity or distance.

---

[2] If the dimension of the feature space $d$ is too high, maximizing the rate reduction may over-estimate the dimension of each class. Hence to learn a good representation, one needs to pre-select a proper dimension for the feature space, as done in the experiments in (Yu et al., 2020).

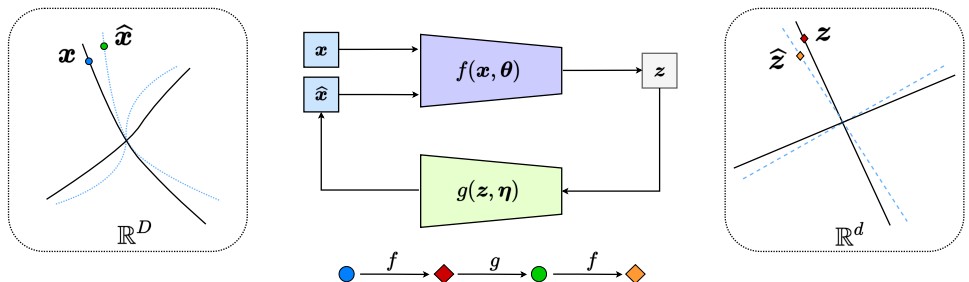

Figure 1: **Closed-loop LDR Transcription.** The encoder $f$ has dual roles: it learns an LDR $z$ for the data $x$ and also discerns any discrepancy in the data $x$ and the decoded $\hat{x}$.

- **Surjectivity:** for all mapped images $z = f(x) \in Z$ of the training data $x \in X$, there are decoded samples $\hat{z} = f(g(z, \eta), \theta) \in \hat{Z}$ close to (ideally the same as) $z$.

## 2.2 MEASURING DISTANCES IN THE FEATURE SPACE AND DATA SPACE

**Contractive Measure.** For the *second* item in the above wishlist, as the representations in the feature space $z$ are by design linear subspaces or (degenerate) Gaussians, we have geometrically or statistically meaningful metrics for both samples and distributions in the feature space $z$. For example, we care about distance between distributions between the features of the original data $Z$ and the transcribed $\hat{Z}$. Since the features of each class, $Z_j$ and $\hat{Z}_j$, are subspace/Gaussian-like, their "distance" can be measured by the rate reduction, with (4) restricted to two sets of equal size):

$$\Delta R(Z_j, \hat{Z}_j) \doteq R(Z_j \cup \hat{Z}_j) - \frac{1}{2}(R(Z_j) + R(\hat{Z}_j)), \tag{6}$$

which, as shown in Fig. 1, measures the space volume between a pair of black lines and blue lines, as per interpretation of rate reduction in (Yu et al., 2020). Then for the "distance" of all, say $k$, classes, we simply sum the rate reduction for all pairs:

$$d(Z, \hat{Z}) \doteq \min_{\eta} \sum_{j=1}^{k} \Delta R(Z_j, \hat{Z}_j) = \min_{\eta} \sum_{j=1}^{k} \Delta R(Z_j, f(g(Z_j, \eta), \theta)), \tag{7}$$

where $Z_j = f(X_j, \theta)$ and $\hat{Z}_j = f(\hat{X}_j, \theta)$. Obviously, a main goal of the learned decoder $g(\cdot, \eta)$ is to *minimize* the distance between these distributions. Notice that if the encoder $f$ preserves (i.e. injective for) the intrinsic structures of the original data $X$,[3] this criterion essentially aims to ensure there will be some decoded sample $\hat{x}$ close to every data sample $x$ – hence the decoder should be "surjective." According to the ideas of IMLE (Li & Malik, 2018), such a requirement could effectively help avoid mode-collapsing or mode-dropping.

**Contrastive Measure.** For the *first* item in our wish-list, however, we normally do not have a natural metric or "distance" for similarity of samples or distributions in the original data space $x$.[4] To alleviate this difficulty, we can measure the similarity or difference between $\hat{X}$ and $X$ through their mapped images $\hat{Z}$ and $Z$ in the feature space (again assuming $f$ is structure-preserving). If we are interested in discerning *any* difference in the distributions of the original and transcribed samples, we may view the MCR$^2$ feature encoder $f(\cdot, \theta)$ as a "discriminator" to *magnify* any difference between all pairs of $X_j$ and $\hat{X}_j$, by simply maximizing, instead of minimizing, the *same quantity* in (7):

$$d(X, \hat{X}) \doteq \max_{\theta} \sum_{j=1}^{k} \Delta R(Z_j, \hat{Z}_j) = \max_{\theta} \sum_{j=1}^{k} \Delta R(f(X_j, \theta), f(\hat{X}_j, \theta)). \tag{8}$$

That is, a "distance" between $X$ and $\hat{X}$ can be measured as the maximally achievable rate reduction between all pairs of classes in these two sets. In a way, this measures how well or bad the decoded $\hat{X}$ aligns with the original data $X$ – hence measuring the goodness of "injectivity". Notice that such a discriminative measure is consistent with the idea of GAN (Goodfellow et al., 2014). Nevertheless, here the MCR$^2$-based discriminator $f$ naturally generalizes to cases when the data distributions are multi-class and multi-modal, and the discriminativeness is measured with a more refined measure – the rate reduction, instead of the typical two-class loss (e.g. cross entropy) used in GANs.

---

[3]This is typically the case for MCR$^2$-based feature representation (Yu et al., 2020).

[4]As mentioned before, finding proper metrics or distance functions on natural images has always been an elusive and challenging task (Wang et al., 2004).

One may wonder the reason why we need the mapping $f(\cdot, \theta)$ to function as a discriminator between $\boldsymbol{X}$ and $\hat{\boldsymbol{X}}$ by maximizing $\max_\theta \Delta R\big(f(\boldsymbol{X}, \theta), f(\hat{\boldsymbol{X}}, \theta)\big)$? Figure 2 gives a simple illustration: there might be many decoders $g$ such that $f \circ g$ is an identity mapping $f \circ g(\boldsymbol{z}) = \boldsymbol{z}$ for all $\boldsymbol{z}$ in the subspace $S_{\boldsymbol{z}}$ in the feature space. However, $g \circ f$ is not necessarily an auto-encoding map for $\boldsymbol{x}$ in the original distribution $S_{\boldsymbol{x}}$ (here for simplicity drawn as a subspace). That is, $g \circ f(S_{\boldsymbol{x}}) \not\subset S_{\boldsymbol{x}}$ (let alone $g \circ f(\boldsymbol{x}) = \boldsymbol{x}$). One should expect, without careful control of the image of $g$, this would typically be the case especially when the support of the distribution of $\boldsymbol{x}$ is extremely low-dimensional in the original high-dimensional data space (e.g. in the millions for images).

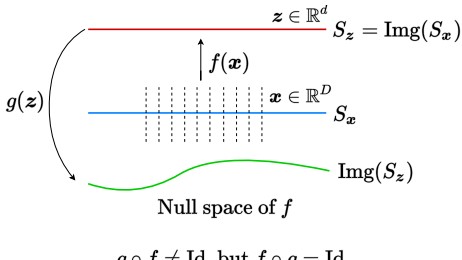

$$g \circ f \neq \mathrm{Id}, \text{ but } f \circ g = \mathrm{Id}$$

Figure 2: $S_{\boldsymbol{x}}$ (red) is the submanifold for the original data $\boldsymbol{x}$; $S_{\boldsymbol{z}}$ (blue) is the image of $S_{\boldsymbol{x}}$ under the mapping $f$, representing the learned feature $\boldsymbol{z}$; and the green curve is the image of the feature $\boldsymbol{z}$ under the decoding mapping $g$.

## 2.3 Encoding and Decoding as a Two-Player MiniMax Game

Comparing (7) and (8), we see the roles of the encoder $f(\cdot, \theta)$ and the decoder $g(\cdot, \eta)$ naturally as a "**a two-player game**": *while the encoder $f$ tries to magnify the difference between the original data and the transcribed; the decoder $g$ aims to minimize the difference.* Now for convenience, let us define the "closed-loop encoding" function:

$$h(\boldsymbol{x}, \theta, \eta) \doteq f\big(g\big(f(\boldsymbol{x}, \theta), \eta\big), \theta\big) : \ \boldsymbol{x} \mapsto \boldsymbol{z}. \tag{9}$$

Ideally, we want this function to be very close to $f(\boldsymbol{x}, \theta)$. With this notation, combining (7) and (8), a closed-loop notion of "distance" between $\boldsymbol{X}$ and $\hat{\boldsymbol{X}}$ can be computed as *an equilibrium point* to the following Min-Max program for the same utility in terms of rate reduction:

$$\mathcal{D}(\boldsymbol{X}, \hat{\boldsymbol{X}}) \doteq \min_\eta \max_\theta \sum_{j=1}^k \Delta R\big(f(\boldsymbol{X}_j, \theta), h(\boldsymbol{X}_j, \theta, \eta)\big). \tag{10}$$

Notice that this only measures the difference between (features of) the original data and its transcribed version. It does not measure how good the representation $\boldsymbol{Z}$ (or $\hat{\boldsymbol{Z}}$) is for the multiple classes within $\boldsymbol{X}$ (or $\hat{\boldsymbol{X}}$). To this end, we may combine the above distance with the original MCR$^2$-type objectives (4): namely, the rate reduction $\Delta R(\boldsymbol{Z})$ and $\Delta R(\hat{\boldsymbol{Z}})$ for the learned LDR $\boldsymbol{Z}$ for $\boldsymbol{X}$ and $\hat{\boldsymbol{Z}}$ for the decoded $\hat{\boldsymbol{X}}$. The overall multi-class Min-Max program for learning the LDR model is:

$$\min_\eta \max_\theta \underbrace{\Delta R\big(f(\boldsymbol{X}, \theta)\big)}_{\text{Expansive encode}} + \underbrace{\Delta R\big(h(\boldsymbol{X}, \theta, \eta)\big)}_{\text{Compressive decode}} + \sum_{j=1}^k \underbrace{\Delta R\big(f(\boldsymbol{X}_j, \theta), h(\boldsymbol{X}_j, \theta, \eta)\big)}_{\text{Contrastive encode \& Contractive decode}} \doteq \mathcal{T}_{\boldsymbol{X}}(\theta, \eta).$$

$$(11)$$

Notice that, without the terms associated with the generative part $h$ or with all such terms fixed as constant, the above objective is precisely the original MCR$^2$ objective.[5] Also, notice that the minimax objective function depends only on the data $\boldsymbol{X}$ hence one can learn the encoder and decoder (parameters) without the need of sampling any other distribution (as needed by GANs)!

As a special case, if $\boldsymbol{X}$ only has one class, the above Min-Max program reduces[6] to a special "binary" form between $\boldsymbol{X}$ and $\hat{\boldsymbol{X}}$:[7]

$$\text{Binary:} \quad \min_\eta \max_\theta \Delta R\big(f(\boldsymbol{X}, \theta), h(\boldsymbol{X}, \theta, \eta)\big). \tag{12}$$

Sometimes, even when $\boldsymbol{X}$ contains multiple classes/modes, one could still view all classes together as one class. Then the above binary objective is to align the union distribution of all classes with their decoded $\hat{\boldsymbol{X}}$. This is typically a simpler task to achieve than the multi-class one (11) since it does not require to learn a more refined multi-class LDR for the data, as we will later see in experiments.

---

[5]In an unsupervised setting, if we view each sample (and its augmentations) as its own class, the above formulation remains exactly the same! The number of classes $k$ is simply the number of independent samples.

[6]as the first two terms become zero.

[7]Notice that this binary case resembles formulation of the original GAN (3) by viewing $\boldsymbol{X}$ and $\hat{\boldsymbol{X}}$ as two classes $\{\boldsymbol{0}, \boldsymbol{1}\}$. Nevertheless, instead of using cross entropy, our formulation adopts a more refined rate reduction measure, which has been shown to promote diversity in the learned representation (Yu et al., 2020).

## 2.4 Discussions about the Minimax Objective

There are many outstanding questions that one may wonder about the above formulation: When does the game $\min_\theta \max_\eta \mathcal{T}_{\boldsymbol{X}}(\theta, \eta)$ have a well-defined Nash-equilibrium point $(\theta^\star, \eta^\star)$? What good characteristics the associated optimal encoder $f^\star$ and $g^\star$ have? For instance, under what conditions, will their composition be or close to be a true auto-encoding $g^\star \circ f^\star \approx$ Id, and will the learned features $\boldsymbol{Z}^\star = f(\boldsymbol{X}, \theta^\star)$ form orthogonal or highly incoherent subspaces?

In this paper, we will not be able to give complete rigorous answers to these questions. The main purpose of this paper is to provide compelling *empirical* evidence that answers to the above questions are highly likely to be *positive* under fair broad conditions (see Section 3). Here we will give some preliminary justification based on characteristics of each team in $\mathcal{T}_{\boldsymbol{X}}(\theta, \eta)$ and will also give reasons why a rigorous characterization of the optimality conditions is mathematically challenging.

Let us first consider maximizing the first two rate reduction terms in $\mathcal{T}_{\boldsymbol{X}}(\theta, \eta)$, assuming $\eta$ fixed and $\boldsymbol{Z}(\theta)$ and $\hat{\boldsymbol{Z}}(\theta)$ are aligned, i.e., $\Delta R(\boldsymbol{Z}_j(\theta), \hat{\boldsymbol{Z}}_j(\theta))$ reaches its minimum 0 for $j \in [k]$ (to be discussed below). Then they essentially maximize the rate reduction $\Delta R(\boldsymbol{Z}(\theta)) + \Delta R(\hat{\boldsymbol{Z}}(\theta))$ for features of the $k$ classes in $\boldsymbol{Z}(\theta) = f(\boldsymbol{X}, \theta)$ and $\hat{\boldsymbol{Z}}(\theta) = h(\boldsymbol{X}, \theta, \eta)$. Based on the results in Yu et al. (2020), a sufficient condition for the rate reduction to reach a (local) maximum is when features of all classes belong to subspaces orthogonal to each other[8]: $\boldsymbol{Z}_i^\top \boldsymbol{Z}_j = \boldsymbol{0}$ and $\hat{\boldsymbol{Z}}_i^\top \hat{\boldsymbol{Z}}_j = \boldsymbol{0}$ for $i \neq j$.

Now let us consider minimizing the $k$ rate reduction terms in the third term of $\mathcal{T}_{\boldsymbol{X}}(\theta, \eta)$, assuming $\theta$ fixed and the $\Delta R(\hat{\boldsymbol{Z}})$ achieves its maximum (as mentioned above). Then, from the property of the $\log \det(\cdot)$ function, we know that each of the $k$ terms in the contractive measure: according to Lemma 10 of Chan et al. (2021), $\Delta R(\boldsymbol{Z}_j, h(\hat{\boldsymbol{Z}}_j(\eta)))$ reaches its minimum (0) when the covariance matrices of $\boldsymbol{Z}_j$ and $\hat{\boldsymbol{Z}}_j$ are the same, i.e., $\boldsymbol{Z}_j \boldsymbol{Z}_j^\top = \hat{\boldsymbol{Z}}_j \hat{\boldsymbol{Z}}_j^\top$. If one further assumes that the encoder is sufficiently injective, perfect alignment of $\boldsymbol{Z}_j$ and $\hat{\boldsymbol{Z}}_j$ suggests perfect alignment of $\boldsymbol{X}_j$ and $\hat{\boldsymbol{X}}_j$.

Based on the above properties, one may show that when the $k$ classes in each of $\boldsymbol{Z}$ and $\hat{\boldsymbol{Z}}$ form $k$ orthogonal subspace and each corresponding pair of $\boldsymbol{Z}_j$ and $\hat{\boldsymbol{Z}}_j$ aligns perfectly, then one reaches a critical point[9] of the minimax game $\min_\theta \max_\eta \mathcal{T}_{\boldsymbol{X}}(\theta, \eta)$. It remains open whether such a critical point is a strict saddle point – a (local) Nash equilibrium.

There are more subtle questions that one could or should ask too, such as: how are the learned features $\boldsymbol{Z}$ and $\hat{\boldsymbol{Z}}$ distributed within the subspaces? Are features for individual sample $\boldsymbol{z}_i \; \hat{\boldsymbol{z}}_i$ (hence $\boldsymbol{x}_i$ and $\hat{\boldsymbol{x}}_i$) well aligned? Results in Yu et al. (2020) suggest that covariance of the features in each subspace needs to be nearly isotropic in order for a rate reduction term to be maximized. When minimizing a rate reduction term, results about the so-called *Brascamp-Lieb inequalities* (Jonathan Bennett et al., 2007) suggest that a necessary condition for the minimum to be reached is when the distribution (density) within each subspace becomes Gaussian. Hence from these results one may conjecture that the optimal distribution in each subspace for the minimax game is Gaussian too. Although theoretically nothing is known about relationships between the features $\boldsymbol{z}_i \; \hat{\boldsymbol{z}}_i$ of individual samples, (multi-class) experiments in next section suggest that they are actually very close.

Despite all these challenging open mathematical questions, one may notice that the aggregated effect of all the terms associated with the decoder $g(\cdot, \eta)$ is precisely "opposite" to the effect of the terms associated with the encoder $f(\cdot, \theta)$. All together, the Min-Max program (11), or (12), aims to strike a good *tradeoff* between maximizing the expressiveness and discriminativeness of the learned representation as well as minimizing any unnecessary cost, in terms of the overall coding rate and errors in decoding, *all measured in rate reduction* (assuming features being subspace Gaussians).

## 3 Empirical Verification on Real-World Imagery Datasets

The experiment section serves three purposes: First, we empirically justify the new formulation by demonstrating good properties of the learned encoder, decoder, and representations. Second, we compare our method with several representative methods from the GAN family and VAE family. Finally, we evaluate the learned LDR through visualization and classification tasks.

---

[8] Here one must assume such a mapping $f$ exists which puts conditions on the original data distribution $\boldsymbol{X}$ and the family of functions that $f(\cdot, \theta)$ can represent.

[9] That is, gradients of the utility $\mathcal{T}$ are zero w.r.t. $\theta$ and $\eta$ because $\frac{\partial \Delta R}{\partial \boldsymbol{Z}}$ all vanishes.

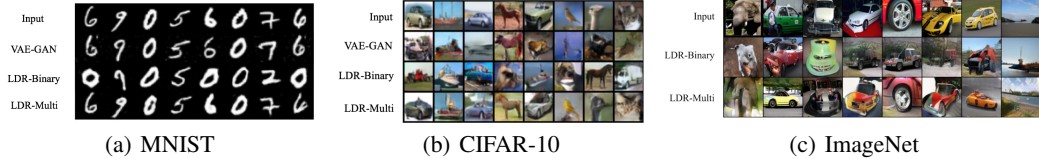

(a) MNIST      (b) CIFAR-10      (c) ImageNet

Figure 3: Qualitative comparison on MNIST, CIFAR-10 and ImageNet. First row: original $X$; Other rows: reconstructed $\hat{X}$ for different methods.

Table 1: Quantitative comparison on MNIST and CIFAR-10. Average Inception scores (IS) (Salimans et al., 2016) and FID scores (Heusel et al., 2017). ↑ means higher is better. ↓ means lower is better.

| Method | | GAN | GAN (LDR-Binary) | VAE-GAN | LDR-Binary | LDR-Multi |
|---|---|---|---|---|---|---|
| MNIST | IS ↑ | 2.08 | 1.95 | **2.21** | 2.02 | 2.07 |
| | FID ↓ | 24.78 | 20.15 | 33.65 | **16.43** | 16.47 |
| CIFAR-10 | IS ↑ | 7.32 | 7.23 | 7.11 | **8.11** | 7.13 |
| | FID ↓ | 26.06 | 22.16 | 43.25 | **19.63** | 23.91 |

**Datasets.** We provide extensive qualitative and quantitative experimental results on the following datasets: MNIST (LeCun et al., 1998), CIFAR-10 (Krizhevsky et al., 2009), STL-10 (Coates et al., 2011), CelebA (Liu et al., 2015), LSUN bedroom (Yu et al., 2015), and ImageNet ILSVRC 2012 (Russakovsky et al., 2015). The network architectures and implementation details can be found in Appendix A.1 and corresponding appendix section for each dataset.

## 3.1 EMPIRICAL JUSTIFICATION OF LDR TRANSCRIPTION

To empirically validate our new framework, we conduct experiments from a small low-variety dataset (MNIST), to a small dataset of diverse real-world objects (CIFAR-10), to higher resolution images (CelebA, LSUN-bedroom), to a large-scale image set (ImageNet). The results are evaluated both quantitatively and qualitatively. Implementation details and more results are given in the Appendix.

**Comparison (IS and FID) with other formulations.** First, we conduct five experiments to fairly compare our formulation with GAN (Radford et al., 2015) and VAE(-GAN) (Larsen et al., 2016) on MNIST and CIFAR-10. Except for the objective function, everything else is exactly the same for all methods (e.g. networks, training data, optimization method). These experiments are: 1). GAN; 2). GAN with its objective replaced by that of the LDR-Binary (12); 3). VAE-GAN ; 4). Binary LDR (12); and 5). Multi-class LDR (11). Some visual comparison is given in Fig. 3. IS (Salimans et al., 2016) and FID (Heusel et al., 2017) scores are summarized in Table 1.

As we see from the above Table 1, replacing cross-entropy with the Equation (12) can improve the generative quality. The two LDR formulations are clearly on par with the others in terms of IS and significantly better in FID. Finally, with the same training datasets, quality of LDR-Multi is lower than LDR-Binary. This is expected as the multi-class task is more challenging. Nevertheless, as we will see soon, images decoded by LDR-Multi align much better with their classes than Binary.

**Visualizing correlation of features $Z$ and decoded features $\hat{Z}$.** We visualize the cosine similarity between $Z$ and $\hat{Z}$ learned from the multi-class objective (11) on MNIST and CIFAR-10, which indicates how close $\hat{z} = f \circ g(z)$ is from $z$. Results in Fig 4 show that $Z$ and $\hat{Z}$ are aligned very well within each class. The block-diagonal patterns for MNIST are sharper than those for CIFAR-10, as images in CIFAR-10 have more diverse visual appearances.

**Visualizing auto-encoding of the data $X$ and the decoded $\hat{X}$.** We compare some representative $X$ and $\hat{X}$ on MNIST, CIFAR-10 and ImageNet (10 classes) to verify how close $\hat{x} = g \circ f(x)$ is to $x$. The results are shown in Fig 5, and visualizations are created from training samples. Visually, the auto-encoded $\hat{x}$ faithfully captures the visual features from its respective training sample $x$. There also exist some auto-encoded images that are almost identical to the original. We refer the reader to Appendix A.2 A.4 A.7 for

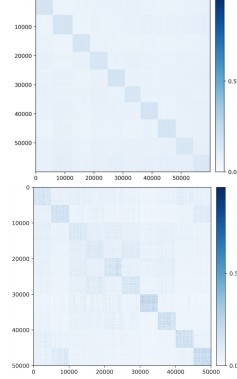

Figure 4: Visualizing the alignment between $Z$ and $\hat{Z}$: $|Z^\top \hat{Z}|$ and in the feature space for MNIST (top) and CIFAR-10 (bottom).

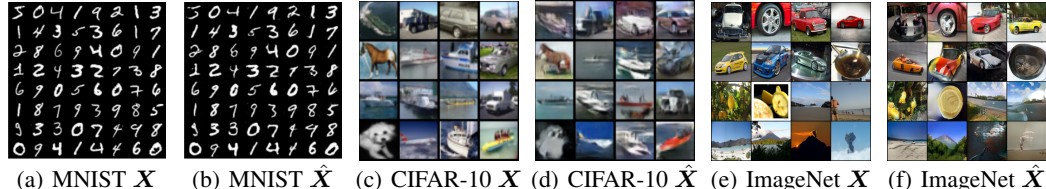

(a) MNIST $\boldsymbol{X}$    (b) MNIST $\hat{\boldsymbol{X}}$    (c) CIFAR-10 $\boldsymbol{X}$    (d) CIFAR-10 $\hat{\boldsymbol{X}}$    (e) ImageNet $\boldsymbol{X}$    (f) ImageNet $\hat{\boldsymbol{X}}$

Figure 5: Visualizing the auto-encoding property of the learned LDR ($\boldsymbol{x} \approx \hat{\boldsymbol{x}} = g \circ f(\boldsymbol{x})$) on MNIST, CIFAR-10, and ImageNet (zoom in for better visualization).

Table 2: Comparison on CIFAR-10 and STL-10. Comparison with more existing methods and on ImageNet can be found in Table 13 in the Appendix.

| Method | | GAN based methods | | | VAE/GAN based methods | | | | |
|---|---|---|---|---|---|---|---|---|---|
| | | SNGAN | CSGAN | LOGAN | VAE-GAN | NVAE | DC-VAE | LDR-Binary | LDR-Multi |
| CIFAR-10 | IS ↑ | 7.4 | 8.1 | **8.7** | 7.4 | - | **8.2** | **8.1** | 7.1 |
| | FID ↓ | 29.3 | 19.6 | **17.7** | 39.8 | 50.8 | **17.9** | 19.6 | 23.9 |
| STL-10 | IS ↑ | 9.1 | - | - | - | - | 8.1 | 8.4 | 7.7 |
| | FID ↓ | 40.1 | - | - | - | - | 41.9 | 38.6 | 45.7 |

more visualizations of results on these datasets, including similar results on transformed MNIST digits and real-life images decoded by randomly sampling the learned feature subspaces.

### 3.2 COMPARISON TO EXISTING GENERATIVE METHODS

Table 2 gives a quantitative comparison of visual quality of our method with others on CIFAR-10, STL-10, and ImageNet. In general, there is a large difference in terms of FID and IS scores between the GAN family and the VAE family of models. SNGAN (Miyato et al., 2018) are commonly used methods in most generative applications while LOGAN (Wu et al., 2019a) is the state-of-the-art method on ImageNet in terms of FID and IS. As we see, even if the rate reduction is not specifically designed nor engineered for visual quality[10], our method is still rather competitive in terms of these metrics.

### 3.3 BENEFITS OF THE LEARNED LDR TRANSCRIPTION MODEL

As we have argued before, the learned LDR transcription model (including the feature $\boldsymbol{z}$, the encoder $f$, and the decoder $g$) can be used for both generative and discriminative purposes.

**Decoding samples from the feature distribution.** Using the CIFAR-10 and CelebA datatsets, we visualize images decoded from samples of learned feature subspace. For the CIFAR-10 dataset, for each class $j$, we first compute the top-4 principal components of the learned features $\boldsymbol{Z}_j$ (via SVD). For each class $j$, we then compute $|\langle \boldsymbol{z}_j^i, \boldsymbol{v}_j^l \rangle|$, the cosine similarity between the $l$-th principal direction $\mathbf{v}_j^l$ and feature sample $\boldsymbol{z}_j^i$. After finding the top-5 $\boldsymbol{z}_j^i$ according to $|\langle \boldsymbol{z}_j^i, \mathbf{v}_j^l \rangle|$ for each class $j$, we reconstruct images $\hat{\boldsymbol{x}}_j^i = g(\boldsymbol{z}_j^i)$. Each row of Fig. 6 is for one principal component. We observe that images in the same row share many common features; images in different rows differ significantly in characteristics like shape, background, and style. See Appendix A.4 for more visualization of principal components learned for all 10 classes of CIFAR-10.

For the CelebA dataset, we calculate the principal components of all learned features in the latent space. Fig 7(a) shows some decoded images along these principal directions. These components seem to clearly disentangle visual attributes/factors such as wearing a hat, changing hair color and wearing glasses. More examples can be found in Appendix A.6. The results are consistent with *the property of MCR² that promotes diversity of the learned features*. Fig. 7(b) shows interpolating features between pairs of training image samples, where for two training images $\boldsymbol{x}_1$ and $\boldsymbol{x}_2$, we reconstruct based on their interpolated feature representations by $\hat{\boldsymbol{x}} = g(\alpha f(\boldsymbol{x}_1) + (1-\alpha) f(\boldsymbol{x}_2)), \alpha \in [0, 1]$. The decoded images show continuous morphing from one sample to another in terms of visual characteristics, as opposed to merely a superposition of the two images.

---

[10]In our current implementation, the original objective is used without any other heuristics or regularization.

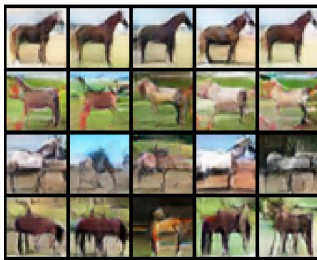 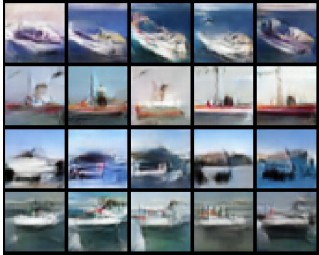

(a) Horse             (b) Ship

Figure 6: **CIFAR-10 dataset.** Visualization of top-5 reconstructed $\hat{x} = g(z)$ based on the closest distance of $z$ to each row (top-4) principal components of data representations for class 7-'Horse' and class 8-'Ship'.

Hat

Hair Color

Glasses

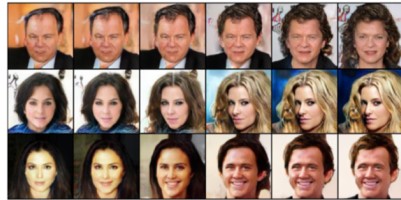

(a) Disentangled attributes as principal components     (b) Interpolation between distinct samples

Figure 7: **CelebA dataset.** (a): Sampling along three principal components that seem to correspond to different visual attributes; (b): Samples decoded by interpolating along the line between features of two distinct samples.

**Encoded features for classification.** Notice that not only is the learned decoder good for generative purpose, but also the encoder is good for discriminative tasks. In this experiment, we evaluate the discriminativeness of the learned LDR model by testing how well the encoded features can help classify the images. We use features of the training images to compute the learned subspaces for all classes, then classify features of the test images based on a simple nearest subspace classifier. Results in Table 3 show that our model gives competitive classification accuracy on MNIST, compared to some of best VAE-based methods. We also tested the classification on CIFAR-10, the accuracy is currently about $80.7\%$. As expected, the representation learned with the multi-class objective is very discriminative and good for classification tasks. This demonstrates the learned LDR model is not only generative but also discriminative.

Table 3: Classification accuracy on MNIST, comparing to classifier based VAE methods (Parmar et al., 2021). Most of those VAE-based methods require auxiliary classifiers to boost classification performance.

| Method | VAE | Factor VAE | Guide-VAE | DC-VAE | LDR-Binary | LDR-Multi |
|---|---|---|---|---|---|---|
| MNIST | 97.12% | 93.65% | 98.51% | 98.71% | 89.12% | 98.30% |

## 3.4 CONCLUSION

This work provides a novel formulation for learning a *both generative and discriminative* representation for multi-class multi-dimensional real-world data. We have provided compelling empirical evidence that the distribution of most data can be effectively mapped to an LDR, a union of incoherent subspaces. The main purpose of this paper is to demonstrate the conceptual simplicity and practical potential of this new representation learning framework, instead of to strive for state of the art performance. Nevertheless, with our preliminary implementation, LDR can be effectively learned for a variety of real-world datasets, from small to large, from domain-specific to diverse. In addition, the so-learned decoder $g$ already enjoys the benefit of GAN for its good generative visual quality and the encoder $f$ with the benefit of AE for its discriminative property. From our experience, the rate reduction based objective can be stably optimized across a wide range of datasets and network architectures without any additional regularizations or engineering tricks. One may notice that there are many ways our method can be significantly improved. For one, in this work, we have simply adopted networks that were designed for GAN or AE, but they may not be optimal for the rate reduction type objectives. Also notice that compared to GAN and AE, our method leads to an *explicit* model for the feature distribution: a mixture of incoherent subspace Gaussians. Such an explicit model has the potential of making many subsequent tasks much easier and better: importance sampling for decoding, classification, or even incremental learning. We leave all these directions, together with all the open mathematical problems, for future investigation.

## ETHICS STATEMENT

All authors agree and will adhere to the conference's Code of Ethics. We do not anticipate any potential ethics issues regarding the research conducted in this work.

## REPRODUCIBILITY STATEMENT

Settings and implementation details of network architectures, optimization methods, and some common hyper-parameters are described in the Appendix A.1. More detailed experimental settings for different datasets can be found in their corresponding Appendix section entitled with the name of the dataset. We will also make our source code available upon request by the reviewers or the area chairs.

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

# A APPENDIX

## A.1 EXPERIMENT SETTINGS AND IMPLEMENTATION DETAILS

**Network backbones.** For MNIST, we use the standard CNN models in Table 4 and Table 5, following the DCGAN architecture (Radford et al., 2015). We resize the MNIST image resolution from $28 \times 28$ to $32 \times 32$ to fit DCGAN architecture. All $\alpha$ in lReLU (lReLU is short for Leaky-ReLU) of the encoder are set to 0.2.

We adopted conv ResNet architectures for CIFAR-10 in Tables 6 and 7, and STL-10 shown in Tables 8 and 9. Each ResBlock up is same as Resnet, but add an up-sampler after the first conv layer. All batch normalization layers of ResBlock in encoder are replaced with spectral normalization layer.

Finally, we use the same architecture for CelebA, LSUN-bedroom, ImageNet-128 (see Tables 10 and 11) as all three datasets have the same $128 \times 128$ resolution. Again, each ResBlock up is same as Resnet, but add an up-sampler after the first conv layer. And all batch normalization layers in encoder are replaced with spectral normalization layer. All experiments utilize this lightweight PyTorch library that provides implementations of popular state-of-the-art GANs and evaluation metrics.

Table 4: Decoder for MNIST.

| $z \in \mathbb{R}^{1 \times 1 \times 128}$ |
| --- |
| $4 \times 4$, stride=1, pad=0 deconv. BN 256 ReLU |
| $4 \times 4$, stride=2, pad=1 deconv. BN 128 ReLU |
| $4 \times 4$, stride=2, pad=1 deconv. BN 64 ReLU |
| $4 \times 4$, stride=2, pad=1 deconv. 1 Tanh |

Table 5: Encoder for MNIST.

| Gray image $x \in \mathbb{R}^{32 \times 32 \times 1}$ |
| --- |
| $4 \times 4$, stride=2, pad=1 conv 64 lReLU |
| $4 \times 4$, stride=2, pad=1 conv 128 lReLU |
| $4 \times 4$, stride=2, pad=1 conv 256 lReLU |
| $4 \times 4$, stride=1, pad=0 conv 128 |

Table 6: Decoder for CIFAR-10.

| $z \in \mathbb{R}^{128}$ |
| --- |
| dense $\rightarrow 4 \times 4 \times 256$ |
| ResBlock up 256 |
| ResBlock up 256 |
| ResBlock up 256 |
| BN, ReLU, $3 \times 3$ conv, 3 Tanh |

Table 7: Encoder for CIFAR-10.

| RGB image $x \in \mathbb{R}^{32 \times 32 \times 3}$ |
| --- |
| ResBlock down 128 |
| ResBlock down 128 |
| ResBlock 128 |
| ResBlock 128 |
| ReLU |
| Global sum pooling |
| dense $\rightarrow 128$ |

Table 8: Decoder for STL-10.

| $z \in \mathbb{R}^{128}$ |
| --- |
| dense $\rightarrow 6 \times 6 \times 512$ |
| ResBlock up 256 |
| ResBlock up 128 |
| ResBlock up 64 |
| BN, ReLU, $3 \times 3$ conv, 3 Tanh |

Table 9: Encoder for STL-10.

| RGB image $x \in \mathbb{R}^{48 \times 48 \times 3}$ |
| --- |
| ResBlock down 64 |
| ResBlock down 128 |
| ResBlock down 256 |
| ResBlock down 512 |
| ResBlock 1024 |
| ReLU |
| Global sum pooling |
| dense $\rightarrow 128$ |

**Training details.** Across all of our experiments, we use Adam (Kingma & Ba, 2014) as our optimizer for all experiments, with hyperparameters $\beta_1 = 0, \beta_2 = 0.9$. The initial value of learning rate is 0.00015 and is scheduled with linear decay. We choose $\epsilon^2 = 0.5$ for both equation 11 and 12 in all LDR experiments. For all LDR-Multi experiments on ImageNet, we only choose 10 classes. The details of the 10 classes as shown in Table 12. Most experiments are trained on RTX 3090ti GPUs.

Table 10: Decoder for CelebA-128, LSUN-bedroom-128, and ImageNet-128.

| $\boldsymbol{z} \in \mathbb{R}^{128}$ |
| :---: |
| dense $\rightarrow 4 \times 4 \times 1024$ |
| ResBlock up 1024 |
| ResBlock up 512 |
| ResBlock up 256 |
| ResBlock up 128 |
| ResBlock up 64 |
| BN, ReLU, $3 \times 3$ conv, 3 Tanh |

Table 11: Encoder for CelebA-128, LSUN-bedroom-128, and ImageNet-128.

| RGB image $\boldsymbol{x} \in \mathbb{R}^{128 \times 128 \times 3}$ |
| :---: |
| ResBlock down 64 |
| ResBlock down 128 |
| ResBlock down 256 |
| ResBlock down 512 |
| ResBlock down 1024 |
| ResBlock 1024 |
| ReLU |
| Global sum pooling |
| dense $\rightarrow 128$ |

Table 12: ID and correspond category for 10 classes of ImageNet

| ID | Category |
| :---: | :---: |
| n02930766 | cab, hack, taxi, taxicab |
| n04596742 | wok |
| n02974003 | car wheel |
| n01491361 | tiger shark, Galeocerdo cuvieri |
| n01514859 | hen |
| n09472597 | volcano |
| n07749582 | lemon |
| n09428293 | seashore, coast, seacoast, sea-coast |
| n02504458 | African elephant, Loxodonta africana |
| n04285008 | sports car, sport car |

## A.2 MNIST

**Settings.** On MNIST dataset, we train our model using DCGAN (Radford et al., 2015) architecture with our proposed models LDR-Multi (11) and LDR-Binary (12). We set the learning rate to $10^{-4}$, batch size to 2048, and training 15,000 iterations. Due to the advantages of the LDR objective, we can achieve between-class discriminative representations while the within-class diversity of these representations can be preserved, which are shown in the following experimental results.

**More results illustrating auto-encoding.** Here we give more reconstruction results, or $\hat{\boldsymbol{X}}$, of our LDR-Multi and LDR-Binary models, compared to their corresponding original input $\boldsymbol{X}$. As shown in the Fig.8, for the LDR-Binary model, it can generate clean digit-like images but the decoded $\hat{\boldsymbol{X}}$ might resemble digits from similar but different classes to the input data $\boldsymbol{X}$ since LDR-Binary tends to only align the distribution of all digits.

In contrast, with the LDR-Multi objective, the decoded $\hat{\boldsymbol{X}}$ not only are coherent with the correct class with the input data $\boldsymbol{X}$, but also show very clear one-to-one mapping between individual sample $\boldsymbol{x}$ and $\hat{\boldsymbol{x}}$ although the objective 11 does not enforce that! Comparing with the results from the VAE-GAN (Larsen et al., 2015), our decoded images preserve much better the individual characteristics of the original samples.

**Images decoded from random samples on the learned multi-class LDR.** Since our LDR-Multi objective function maps input data of each class into a different (orthogonal) subspace in the feature space, we can generate images conditioned on each class by random sampling $\boldsymbol{z}$ in the subspace of each class and then decode them back to the input space as $\hat{\boldsymbol{x}}$.

To do random sampling in the learned subspace, we first calculate the mean feature $\bar{\boldsymbol{z}}_j$ and the singular vectors $\mathbf{v}_j^i$ of the SVD (or principal components) of the learned features $\boldsymbol{Z}_j$ of the class $j$ of the training data, where index $i$ represents the $i$th principal components. We only use top $r = 8$ principal

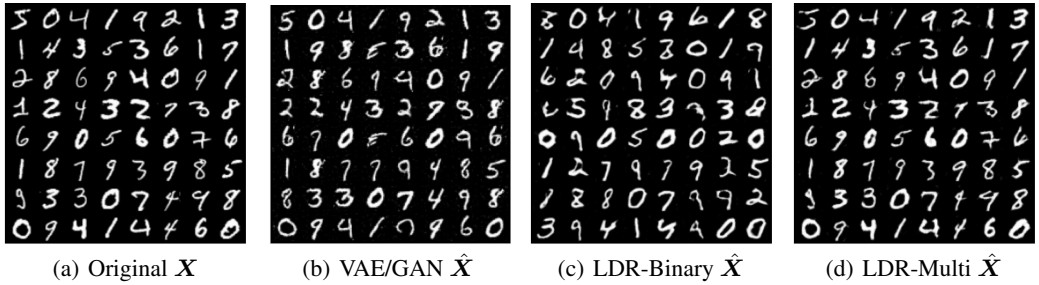

(a) Original $\boldsymbol{X}$     (b) VAE/GAN $\hat{\boldsymbol{X}}$     (c) LDR-Binary $\hat{\boldsymbol{X}}$     (d) LDR-Multi $\hat{\boldsymbol{X}}$

Figure 8: The comparison of the reconstruction results of different methods with the input data.

components of each class on MNIST dataset. These statistics of the subspace can be used for guiding the random sampling. Then we sample $\boldsymbol{z}$ randomly along the principal components and around the mean feature as

$$\boldsymbol{z}_{random\_j} = \bar{\boldsymbol{z}}_j + \alpha \sum_{i=1}^{r} n_i * \sigma_j^i * \mathbf{v}_j^i, \tag{13}$$

where $\bar{\boldsymbol{z}}_j$ is the mean feature of class $j$, $\sigma_j^i$ and $\mathbf{v}_j^i$ are the $i$-th singular value and principal component of class $j$, $n_i$ are i.i.d. Gaussian $\mathcal{N}(0, 1)$ random variables. That is, the feature in each subspace/class is modeled by a $r$-dimensional multivariate Gaussian, with variances $\sigma_j^i$ which characterize variances of the training data in the feature space. Here, $\alpha$ is a hyper-parameter that controls the sampling range. As for visualization of random generated images $g(\boldsymbol{z}_{random\_j})$ conditioned on the given class, we compare our method with some other conditional generation method such as ACGAN (Odena et al., 2017) and InfoGAN (Chen et al., 2016) (For ACGAN and InfoGAN, we generate images conditioned on class labels with randomly sampled latent $\boldsymbol{z}$ according the procedures mentioned in their respective works). Our model can give realistic and correct conditional generation results with high diversity in each class, while other methods may make mistakes in the generation between some similar classes such as classes 3 and 5 for InfoGAN.

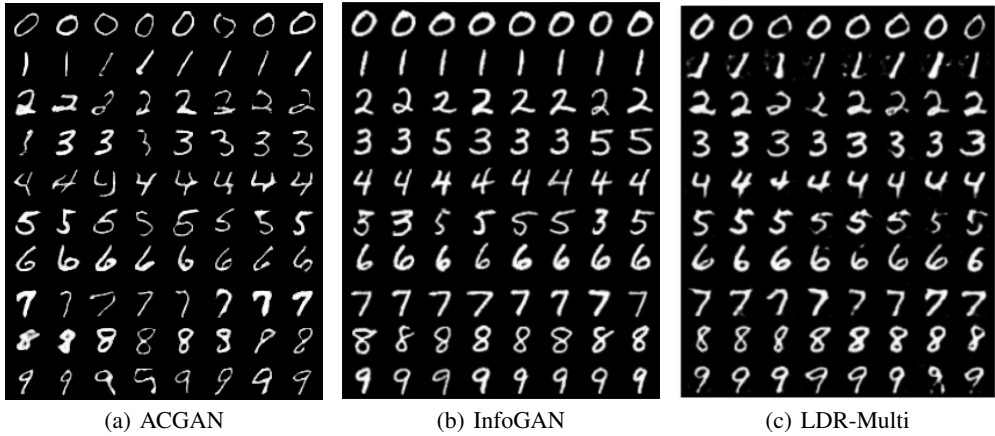

(a) ACGAN     (b) InfoGAN     (c) LDR-Multi

Figure 9: Comparison of randomly generated images conditioned on each class.

### A.3 TRANSFORMED MNIST

**Settings.** In this experiment, we verify that our LDR-Multi model can preserve diverse data modes in the learned feature embeddings. We construct a transformed MNIST dataset with 5 modes: normal, large(1.5 ×), small(0.5 ×), rotate $45°$ left, and rotate $45°$ right. Each image data will be randomly transformed to one of the modes. Representative examples of such training data can be found in Figure 10(a). We train the model with learning rate 1e-4 and batch size 2048 for 15,000 iterations.

**Auto-encoding results.** Figure 10(b) gives the decoded results of the training data with different modes. Even though the data are now much more diverse for each class, decoder learned from the LDR-Multi objective can still achieve high sample-wise similarity to the original images.

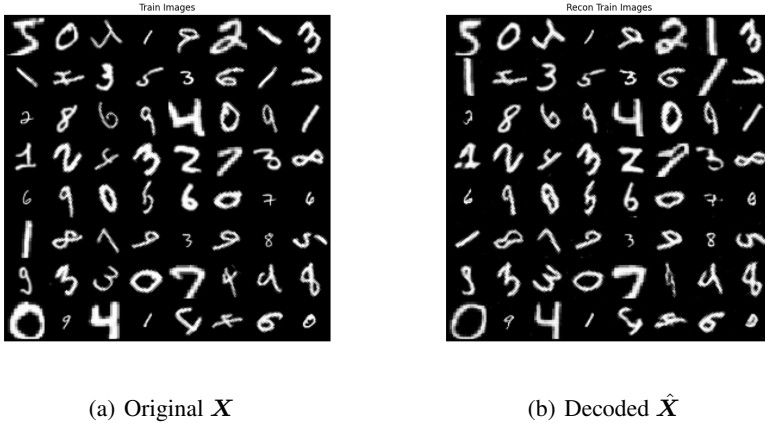

(a) Original $\boldsymbol{X}$                    (b) Decoded $\hat{\boldsymbol{X}}$

Figure 10: Original (training) data $\boldsymbol{X}$ and their decoded version $\hat{\boldsymbol{X}}$ on transformed MNIST.

**Identifying different modes.** Similar to the earlier experiments of Fig. 6 for CIFAR-10 in the main paper, we find the top principal components of features of each class $\boldsymbol{Z}_j$ (via SVD) and generate new images using the learned decoder $g$ from features of the training images aligned the best with these components.

In Fig 11, we select three classes 0, 1, 2 and visualize samples from top $r = 8$ principal components for each class. Each row represents one principal components direction. As it can be seen in the figure, the decoded images along each principal component shows similar mode and the modes along different component directions are rather incoherent. All major modes of the original data can be identified as one of these principal directions. This clearly shows that our LDR-Multi model can keep the different modes within each class of the data $\boldsymbol{X}_j$ as the principal component directions of $\boldsymbol{Z}_j$, and these modes can also be retained in the decoded images $\hat{\boldsymbol{X}}_j$.

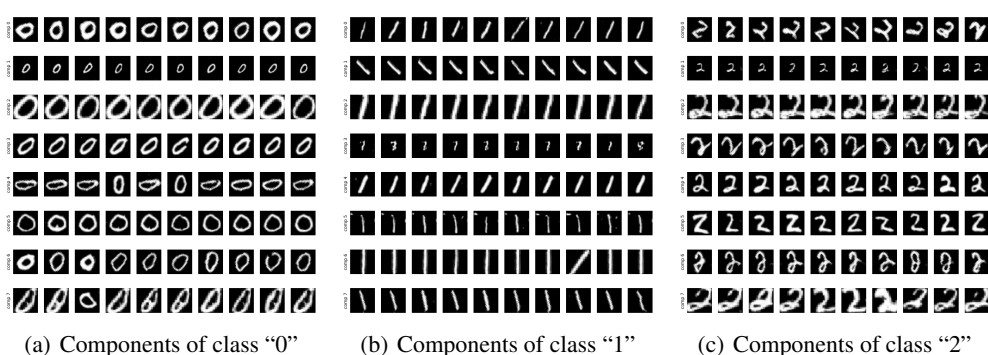

(a) Components of class "0"      (b) Components of class "1"      (c) Components of class "2"

Figure 11: The reconstructed images $\hat{\boldsymbol{X}}$ from the features $\boldsymbol{Z}$ best aligned along top-8 principal components on the transformed MNIST dataset. Each row represents a different principal component.

## A.4   CIFAR-10

**Settings.** For all experiments on CIFAR-10, we follow the common training hyper-parameters in section A.1. Beyond that, for each experiment, we run 450,000 iterations with mini-batch size 1600.

**Images decoded from random samples on the learned multi-class LDR.** We sample $z$ in the feature space randomly along the principal components and around the mean feature of each class $Z_j$ as in the MNIST case, according to equation (13). The generated images from the sampled features are illustrated in Fig 12, one row per class. As we see, the generator learned from the LDR-multi objective is capable of generating diverse images for each class.

Further, for visualization of random generated images $g(z_{random\_j})$ conditioned on the given class, we also compare our method with some other conditional generation method such as ACGAN (Odena et al., 2017) and InfoGAN (Chen et al., 2016). For all three experiments, we have randomly sampled 8 images per class in CIFAR-10. For more complex dataset like CIFAR-10, our model can give more realistic conditional generation results for different classes with high diversity within each class.

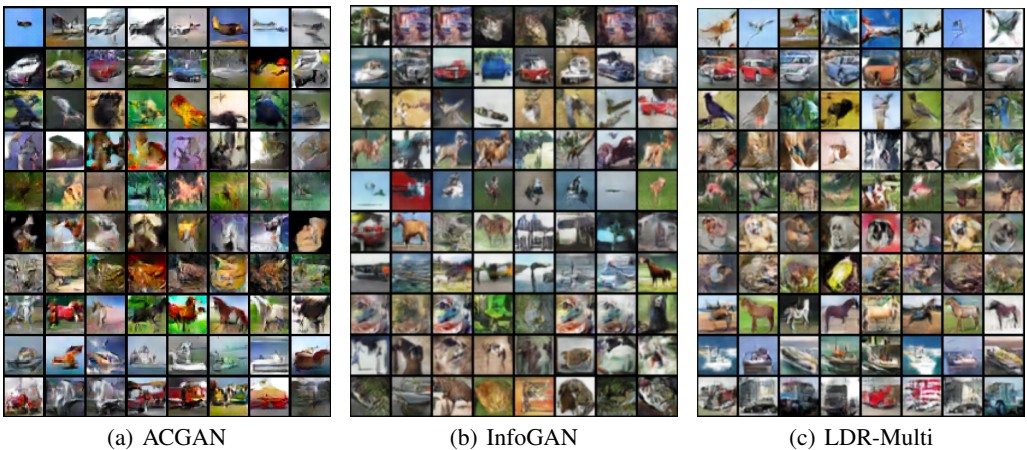

|     (a) ACGAN     |     (b) InfoGAN     |     (c) LDR-Multi     |

Figure 12: Comparison of randomly generated images conditioned on each class.

**Generating image along different PCA components for each class.** For each class, we first compute top-10 principal components (singular vectors of the SVD) of $Z$ and then for each of the top singular vectors, we display in each row the top-10 reconstructed image $\hat{X}$ whose $Z$ are closest to the singular vector using method described in the main body of the paper, Section 3.3. The results are given in Figure 13 below.

### A.5  STL-10

**Settings.**   For all experiments on STL-10, we follow the common training hyper-parameters in section A.1. For LDR-Binary setting, we train 150,000 iterations. For LDR-Multi setting, we initialize the weights from the 20,000-th iteration of LDR-Binary checkpoint and train for another 80,000 iterations (with the LDR-Multi objective). The IS and FID scores on the STL-10 dataset are reported in Table 13, on par or even better than existing methods such as SNGAN (Miyato et al., 2018) or DC-VAE (Parmar et al., 2021).

**Visualizing auto-encoding property for LDR-Binary.** We visualize the original images $x$ and their decoded $\hat{x}$ using the LDR model learned from LDR-Binary objective. The results are shown in Figure 14 for STL-10.

### A.6  CELEB-A AND LSUN

To verify that our formulation works on images of higher-resolution, we conduct experiments on the Celeb-A and LSUN datasets, which have a resolution of $128 \times 128$.

**Settings.**   For all experiments on these datasets, we follow the common training hyper-parameters in Section A.1. We choose a 300 mini-batch size for Celeb-A and LSUN. Both of them are trained with the LDR-Binary formulation, and for 450,000 iterations.

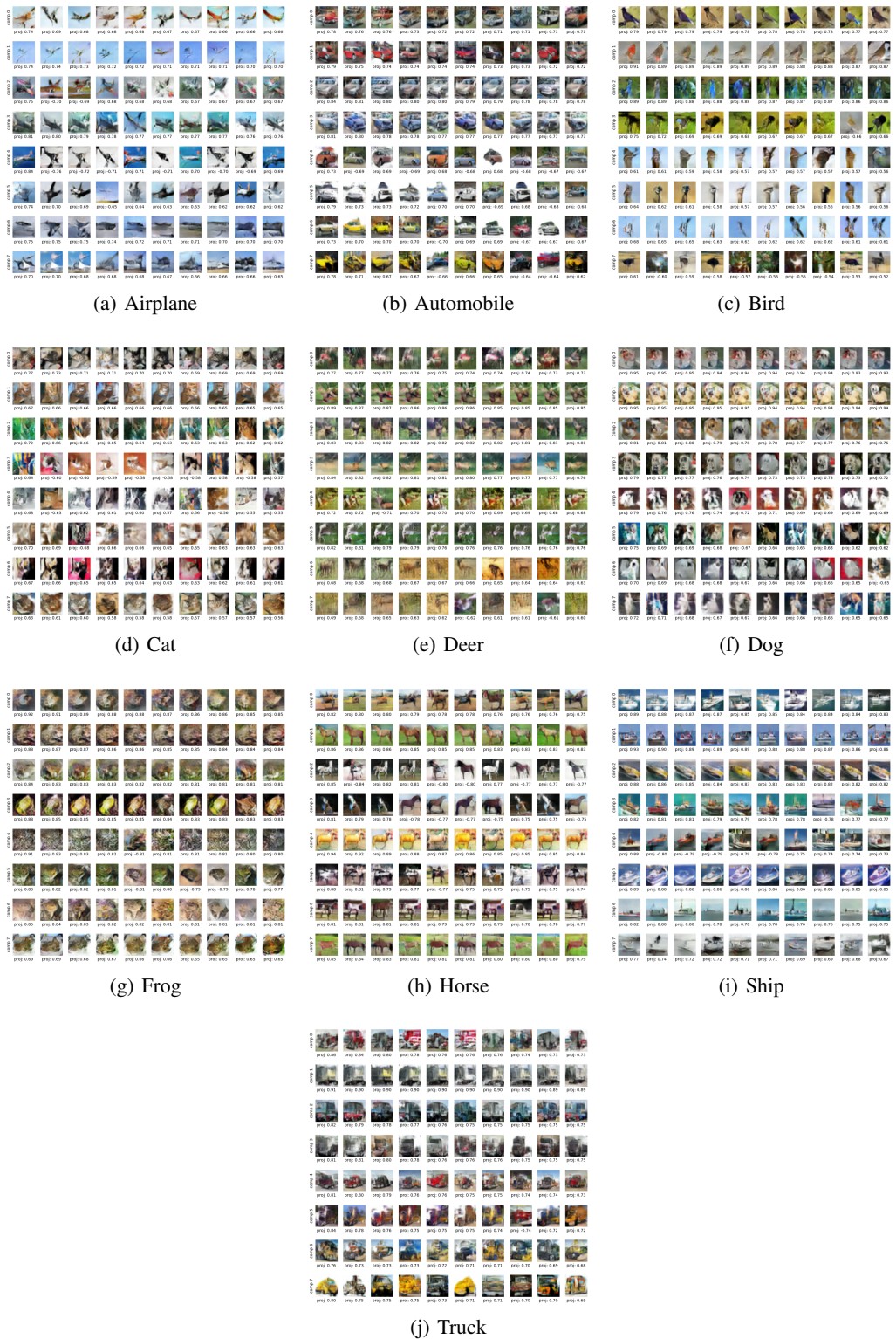

Figure 13: Reconstructed images $\hat{X}$ from features $Z$ close to the principal components learned for the 10 classes of CIFAR-10.

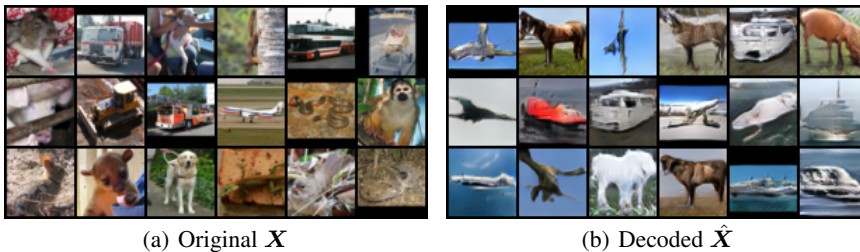

(a) Original $\boldsymbol{X}$           (b) Decoded $\hat{\boldsymbol{X}}$

Figure 14: Visualizing the original $\boldsymbol{x}$ and corresponding decoded $\hat{\boldsymbol{x}}$ results on STL-10 dataset. Note the model is trained from LDR-Binary (12) hence sample or class wise correspondence is relatively poor. But the decoded image quality is very good.

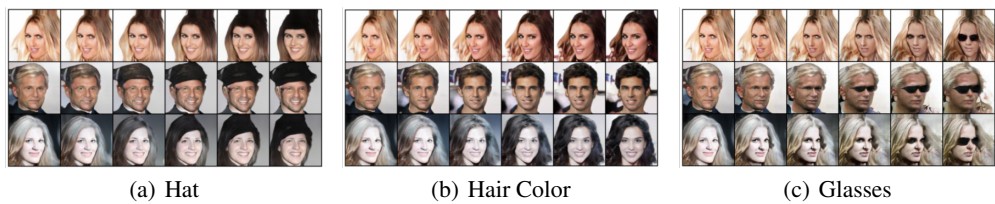

(a) Hat          (b) Hair Color          (c) Glasses

Figure 15: Sampling along the 9-th, 19-th, and 23-th principal components of the learned features $\boldsymbol{Z}$ seems to manipulate the visual attributes for generated images, on the CelebA dataset.

**Generating image along different PCA components.** We calculate the principal components of the learned features $Z$ in the latent subspace. We manually choose 3 principle components which are related to hat, hair color, and glasses (see Fig 15). The three components are 9-th, 19-th, and 23-th respectively from the overall 128 principal components. These principal directions seem to clearly disentangle visual attributes/factors such as wearing hat, changing hair color, and wearing glasses.

**Images generated from random sampling of the feature space.** We sample $\boldsymbol{z}$ randomly according to the following Gaussian model:

$$\boldsymbol{z}_{random} = \bar{\boldsymbol{z}} + \alpha \sum_{i=1}^{r} n_i * \sigma_i * \mathbf{v}_i, \tag{14}$$

where $\bar{\boldsymbol{z}}$ is the mean feature, $\sigma_i$ and $\mathbf{v}_i$ are the $i$th singular value and singular vector, $n_i$ are i.i.d. Gaussian $\mathcal{N}(0,1)$ random variables. As before $\alpha$ is a hyper-parameter to control the sampling range. We use top r=100 principle components for random sampling. The random generated images are realistic and diverse. (see Fig 16)

**Visualizing auto-encoding property for LDR-Binary.** We visualize the original image $\boldsymbol{x}$ and their decoded $\hat{\boldsymbol{x}}$ using the LDR model learned from LDR-Binary formulation. The results are shown in Fig 17 and Fig 18 for the Celeb-A dataset and the LSUN dataset, respectively. The LDR-Binary formulation can give very good visual quality for $\hat{\boldsymbol{x}}$ but cannot ensure sample to sample alignment. Nevertheless, the decoded $\hat{\boldsymbol{x}}$ seems to be similar in style to the original $\boldsymbol{x}$. We believe it manages to align only the dominant principal component(s) associated with the most salient visual attributes, say pose of the face for Celeb-A or layout of the room for LSUN, between features of $\boldsymbol{X}$ and $\hat{\boldsymbol{X}}$.

## A.7 IMAGENET

**Settings.** To verify that our formulation works on large-scale datasets, we train a model on the entire ImageNet. For all experiments on the ImageNet, we follow the common training hyper-parameters in Section A.1.

We first train our model with LDR-Binary (12) with a mini-batch size 1800 on the whole ImageNet ILSVRC 2012 dataset. The number of training iterations is 450,000.

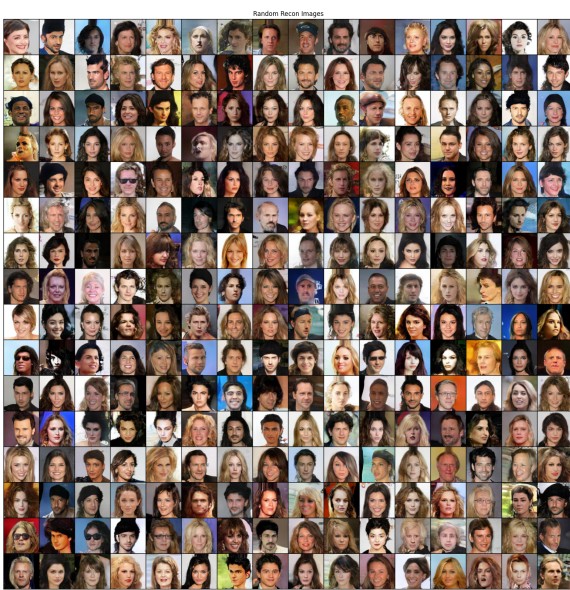

Figure 16: Images decoded from randomly sampled features, as a learned Gaussian distribution (14), for the CelebA dataset.

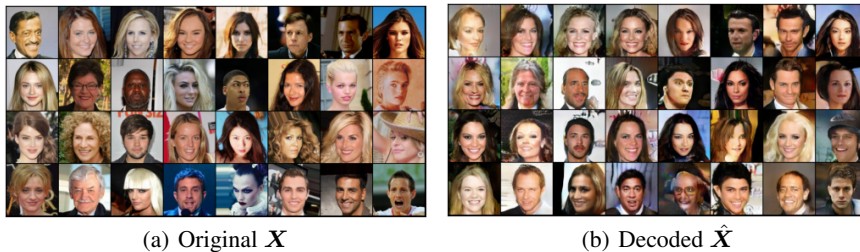

(a) Original $\boldsymbol{X}$        (b) Decoded $\hat{\boldsymbol{X}}$

Figure 17: Visualizing the original $\boldsymbol{x}$ and corresponding decoded $\hat{\boldsymbol{x}}$ results on Celeb-A dataset. The LDR model is trained from LDR-Binary (12).

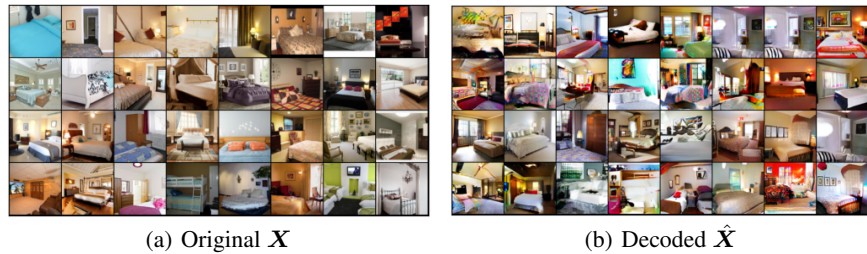

(a) Original $\boldsymbol{X}$        (b) Decoded $\hat{\boldsymbol{X}}$

Figure 18: Visualizing the original $\boldsymbol{x}$ and corresponding decoded $\hat{\boldsymbol{x}}$ results on LSUN-bedroom dataset. The LDR model is trained from LDR-Binary (12).

Table 13: Comparison on CIFAR-10, STL-10, and ImageNet.

| Method | CIFAR-10 | | STL-10 | | ImageNet | |
|---|---|---|---|---|---|---|
| | IS↑ | FID↓ | IS↑ | FID↓ | IS↑ | FID↓ |
| *GAN based methods* | | | | | | |
| DCGAN Radford et al. (2015) | 6.6 | - | 7.8 | - | - | - |
| SNGAN Miyato et al. (2018) | 7.4 | 29.3 | **9.1** | **40.1** | - | 48.73 |
| CSGAN Wu et al. (2019b) | 8.1 | 19.6 | - | - | - | - |
| LOGAN Wu et al. (2019a) | **8.7** | **17.7** | - | - | - | - |
| *VAE/GAN based methods* | | | | | | |
| VAE Kingma & Welling (2013) | 3.8 | 115.8 | - | - | - | - |
| VAE/GAN Larsen et al. (2016) | 7.4 | 39.8 | - | - | - | - |
| NVAE Vahdat & Kautz (2020) | - | 50.8 | - | - | - | - |
| DC-VAE Parmar et al. (2021) | **8.2** | **17.9** | 8.1 | 41.9 | - | - |
| LDR-Binary (ours) | **8.1** | **19.6** | 8.4 | 38.6 | 7.74 | **46.95** |
| LDR-Multi (ours) | 7.1 | 23.9 | 7.7 | 45.7 | 6.44 | 55.51 |

After that, we fine-tune the Binary-pretrained model with LDR-Multi (11), on 10 selected classes. Information about the 10 classes can be found in Table 12. The fine-tune mini-batch size is 1024, and we train another 35,000 iterations. This experiment took 8 A100-SXM4 GPUs, each with 40GB of CUDA memory for 120 GPU hours. Note that our choice of mini-batch size is substantially larger than those commonly adopted in other works while training on the ImageNet (e.g. 128 in Miyato et al. (2018)). We empirically observe that training with a larger mini-batch generates images of better quality and clearer class alignment. This is consistent with the proposed LDR framework as the LDR-Multi objective explicitly encourages alignment of class distributions, therefore benefiting from a larger batch that better captures overall data distributions. We leave a more rigorous study of the effect of batch size for future work.

Due to the heavy computation of such large batch size, we present the following intermediate results obtained at 35,000 iterations whereas most existing methods run with magnitude larger number of iterations. Nevertheless, these intermediate results already verify the efficacy of our framework. In addition, we present the full version of the comparison with existing generative methods in Table 13. We see the SI and FID scores for LDR-Multi degraded a little after the finetuning. This is expected as learning a more refined separation and alignment of 10 classes is a more challenging task than 2 classes. This is consistently observed from experiments on other datasets too.

**Visualizing feature similarity for LDR-Multi.** We visualize the cosine similarity among features $\boldsymbol{Z}$ of different classes learned from the LDR-Multi objective in Figure 19. In addition, we provide the visualization of alignment between features $\boldsymbol{Z}$ and decoded features features $\hat{\boldsymbol{Z}}$. These results demonstrate that not only the encoder has already learnt to discriminate between classes, $\boldsymbol{Z}$ and $\hat{\boldsymbol{Z}}$ also are aligned clearly within each class.

**Visualizing auto-encoding property for LDR-Multi.** We visualize the original image $\boldsymbol{x}$ and their decoded $\hat{\boldsymbol{x}}$ using the LDR model fine-tuned using LDR-Multi formulation. The results are shown in Fig 20 for the selected 10 classes in ImageNet. The LDR-Multi formulation can give good visual quality for $\hat{\boldsymbol{x}}$ as well as decent sample-to-sample alignment.

## A.8   ABLATION STUDY ON OBJECTIVE FUNCTIONS

To empirically validate the importance of the rate reduction ($\Delta R$) loss, we keep the closed-loop formulation but replace all rate reduction ($\Delta R$) loss terms in (11) with corresponding cross-entropy loss.

To replace the rate reduction ($\Delta R$) terms in the objective function (11) with cross-entropy, we introduce a linear mapping $\boldsymbol{W} \in \mathbb{R}^{d \times k}$ to map $\boldsymbol{Z} \in \mathbb{R}^{d \times n}$ from feature space to logits $\gamma = \boldsymbol{Z}^\top \boldsymbol{W}$. We then calculate the softmax cross-entropy function on logits $\gamma$ and one hot label matrix $\boldsymbol{Y}$. Here $\mathcal{H}(\gamma, \boldsymbol{Y}) = \sum_{i=1}^{n} \sum_{j=1}^{k} Y_{ij} \log \frac{e^{\gamma_{ij}}}{\sum_{j=1}^{k} e^{\gamma_{ij}}}$ is the formulation of softmax cross-entropy function and

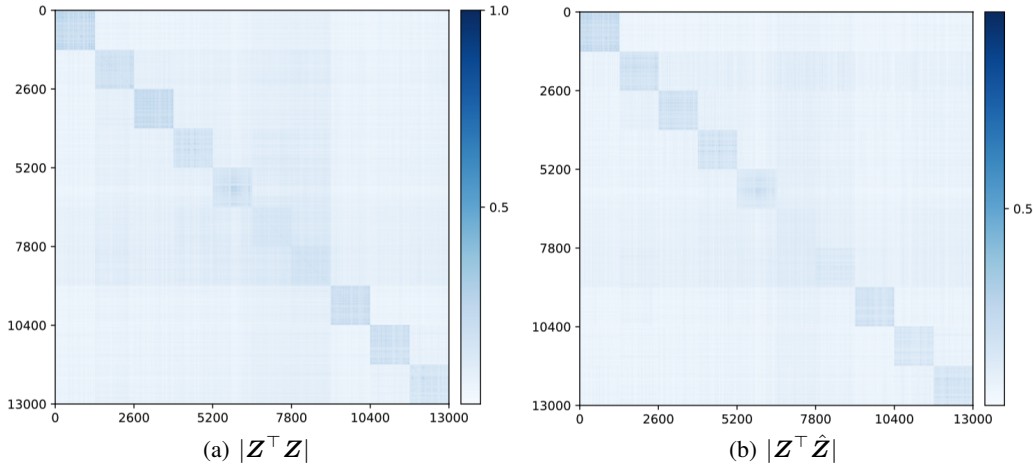

Figure 19: Visualizing feature alignment: (a) among features $|\boldsymbol{Z}^\top \boldsymbol{Z}|$, (b) between features and decoded features $|\boldsymbol{Z}^\top \hat{\boldsymbol{Z}}|$. These results obtained after 200,000 iterations.

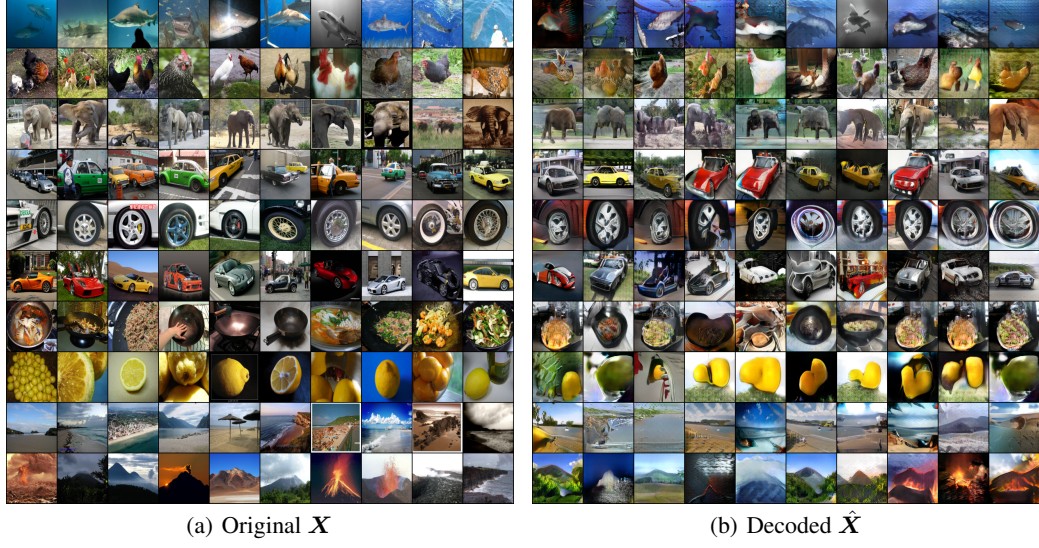

Figure 20: Visualizing the original $\boldsymbol{x}$ and corresponding decoded $\hat{\boldsymbol{x}}$ results on ImageNet (10 classes). The LDR model fine-tuned using LDR-Multi (11).

$\boldsymbol{Y} \in \mathbb{R}^{n \times k}$ is one hot label matrix. Then, we can replace the first two terms of (11) ($\Delta R(\boldsymbol{Z})$ and $\Delta R(\hat{\boldsymbol{Z}})$) with $\mathcal{H}(\boldsymbol{Z}^\top \boldsymbol{W}, \boldsymbol{Y})$ and $\mathcal{H}(\hat{\boldsymbol{Z}}^\top \boldsymbol{W}, \boldsymbol{Y})$. For the third term of (11), we extract $j$-th class one hot feature $\gamma_j = \boldsymbol{Z}_j^\top \boldsymbol{W}$, $\hat{\gamma}_j = \hat{\boldsymbol{Z}}_j^\top \boldsymbol{W}$ from $\boldsymbol{Z}$ and $\hat{\boldsymbol{Z}}$, and define the distance $\mathcal{D}(\gamma_j, \hat{\gamma}_j) = \frac{e^{\gamma_j}}{e^{\gamma_j} + e^{\hat{\gamma}_j}}$ of them. For the third term of (11), we further introduce $k$ linear layers as discriminators $\{\mathcal{D}_j\}_{j=1}^k$ for each class. Then, we replace the third term with the GAN's objective function as $\sum_{j=1}^k \mathbb{E}[\log \mathcal{D}_j(\boldsymbol{Z}_j)] + \mathbb{E}[\log(1 - \mathcal{D}_j(\hat{\boldsymbol{Z}}_j))]$ [11]. Now, we have the cross-entropy version objective function (15) for closed loop framework. We denote the closed loop framework with cross-entropy as Closed-loop-CE.

---

[11] $\mathbb{E}[\boldsymbol{X}]$ denote the expectation of $\boldsymbol{X}$.

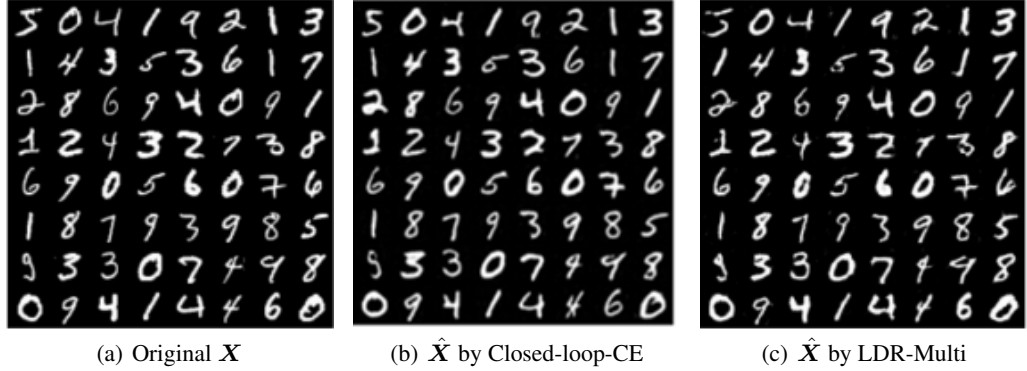

(a) Original $\boldsymbol{X}$      (b) $\hat{\boldsymbol{X}}$ by Closed-loop-CE      (c) $\hat{\boldsymbol{X}}$ by LDR-Multi

Figure 21: The comparison of sample-wise reconstruction between Close-loop-CE and LDR-Multi.

$$\min_{\eta} \max_{\theta, \boldsymbol{W}, \mathcal{D}} \mathcal{T}_{\boldsymbol{X}}(\theta, \eta, \boldsymbol{W}, \mathcal{D}) \doteq \mathcal{H}(\boldsymbol{Z}^{\top}\boldsymbol{W}, \boldsymbol{Y}) + \mathcal{H}(\hat{\boldsymbol{Z}}^{\top}\boldsymbol{W}, \boldsymbol{Y}) +$$

$$\sum_{j=1}^{k} \mathbb{E}[\log \mathcal{D}_j(\boldsymbol{Z}_j)] + \mathbb{E}[\log(1 - \mathcal{D}_j(\hat{\boldsymbol{Z}}_j))]. \tag{15}$$

We run the experiments on MNIST and CIFAR10. The architectures of MNIST and CIFAR10 are given in Table 4 to Table 7[12].

**Results on MNIST.** The training hyper-parameters of LDR-Multi and Closed-loop-CE on MNIST are following Section A.1. Comparisons between LDR-Multi and Closed-loop-CE are listed in Figure 21, 22, and 23.

Figure 21(b) and 21(c) show the reconstructed images $\hat{\boldsymbol{X}}$ from Closed-loop-CE and LDR-Multi. Both methods can give sample-wise reconstruction results due to the transcription framework. However, comparing training images whose features are best aligned with the principal components of class '2' in Figure 22, we see that the principal components of CE features do not correspond to consistent visual attributes of the images, whereas ours do.

From the heatmaps in Figure 23(a) and 23(b), we see the features learned by rate reduction possess clear orthogonal subspace structures, whereas those learned by Closed-loop-CE do not. Moreover, Figure 23(c) and 23(d) show that the learned features of LDR-Multi have higher singular values for the top principal components of each class, corresponding to a more linearized and diverse feature distribution, whereas those by Closed-loop-CE do not.

**Failed Attempts on CIFAR-10 with Cross Entropy.** The training hyper-parameters of Closed-loop-CE on CIFAR10 follow Section A.1. We do the grid search on three hyper-parameters: learning rate $\{1.5 \times 10^{-2}, 1.5 \times 10^{-3}, 1.5 \times 10^{-4}\}$, mini batch size (800 or 1600), and inner loop (1,2,3,4), conduct 24 experiments in total. All cases of the Closed-loop-CE fail to converge or experience model collapse on the CIFAR-10 dataset.

---

[12]In the context of this section, we use the term Decoder and Generator interchangeably; similarly for Encoder and Discriminator.

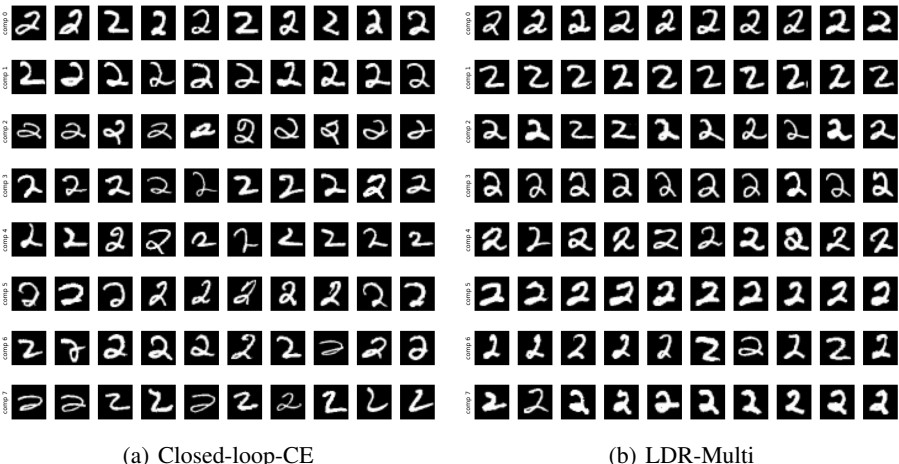

(a) Closed-loop-CE

(b) LDR-Multi

Figure 22: Training samples along different principal components of the learned features of digit '2'.

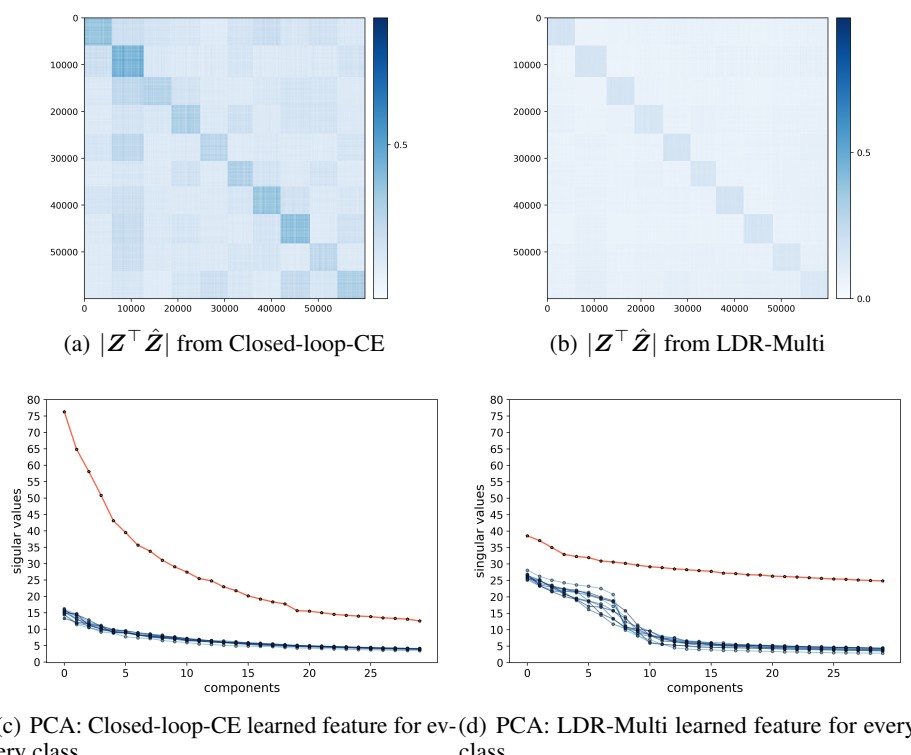

(a) $|\boldsymbol{Z}^\top \hat{\boldsymbol{Z}}|$ from Closed-loop-CE

(b) $|\boldsymbol{Z}^\top \hat{\boldsymbol{Z}}|$ from LDR-Multi

(c) PCA: Closed-loop-CE learned feature for every class

(d) PCA: LDR-Multi learned feature for every class

Figure 23: Comparison Closed-loop-CE and LDR-Multi on $|\boldsymbol{Z}^\top \hat{\boldsymbol{Z}}|$ and PCA singular values.

