# OpenReview forum: "Closed-Loop Data Transcription to an LDR via Minimaxing Rate Reduction"
_ICLR.cc/2022/Conference — ICLR 2022 Submitted_

### Official Review · Reviewer_hk2J · 2021-11-02

**Correctness:** 3
**Technical Novelty And Significance:** 3
**Empirical Novelty And Significance:** 3
**Recommendation:** 3
**Confidence:** 3

**Main Review:**

Strengths.
- the core idea of the paper is novel as far as I can tell. I'd argue it has some resemblance to CycleGan (Zhu et al. 2017), but the setting on top of rate reduction and motivation is significantly different.
- the experiments demonstrate that the method can work well as a generative method, even though it formulates its losses only in latent space.

Weaknesses.
- The paper is not as self-contained as it could be, i.e. it is near impossible to understand without reading the core referred literature first. It would help readability a lot if core intuitions and concepts would be briefly discussed when introduced. E.g. when introducing MCR2 in equation 4, it would help the reader a lot what the target of each of the two terms is. Another example, after equation 6 it is stated that this measures the volume - why? (if this is not easy to explain, then there should be a reference to where this is explained). The math also needs more clarification, for example what is the union in equation 6 (Z and \hat Z are both \in Rˆ{dxn}, or)? Is the \Delta R in equation 4 and equation 6 really the same (or is it semantically overloaded)?  writing (and logic of the writing) needs to be worked on.
- Experiments. The comparisons on generative models in Table 1 are done with rather old baselines (all being at least 4 years old). E.g. DCGAN (Radford 2015) could be replaced by StyleGAN2/3 etc (similar for the VAE methods). It should also be reported what the training time for the method on the datasets is, and how it scales with dataset size and resolution. Also, it should be discussed that the model only roughly encodes the semantics, and sometimes disregards even color (e.g. as seen in Figure 5 where the red car turns yellow). In Figure 14 in the abstract it can be seen that the the model seems to sometimes collapse inputs into the same output? If so this should be discussed.
- Already in the Abstract it is claimed that the model learns a discriminative representation, and it is shown in Table 3 to perform in the ballpark of VAE methods on MNIST data. As the model is also trained on ImageNet (see Fig5), why is no discrimnative comparison performed on ImageNet? It'd be interesting to see if the model can scale up to more classes and more complex datasets (what is the scaling behaviour theoretically of the model in the latent space? Does the dimensionality need to be proportional to the number of classes to ensure the possibility of orthogonality?) .

Minor Issues.
- The Introduction should start with a section contextualizing the work, i.e. review of context and background, what has been done by others broadly in the field that led to the current work.
- Equation 2,3 etc. - the result of function g is not a tuple (X, \hat X). The figure should be changed.
- After eq. 3 it is written "later studies [...] surrogate to earth mover's distance". I think this is wrong. The original GAN paper showed that GAN's minimize Jensen-Shannon distance, and Arjovsky et al showed how to build the WGAN that operates on earth mover's distance.
- Also on pg 2. - Combination of AE and GAN: ".. started with somewhat different motivation, they have evolved...". While true that both methods are quite different, as both are generative models, it was this motivation (generative modeling) that they become successful in modeling of real-world data. So, the logic behind this sentence is confusing.


**Summary Of The Paper:**

The paper "Closed-Loop Data Transcription To An LDR via Minimaxing Rate Reduction" introduces a two-player minimax game between an encoder and a decoder to yield a linear discriminative representation (LDR). It achieves this by building on top of the recently proposed MCR2 rate reduction principle, and then motivates a contractive and contrastive measure to yield a minimax game. The paper shows generative empirical results on MNIST, CIFAR-10 and ImageNet, as well as discriminative classification accuracies on MNIST.

**Summary Of The Review:**

While the paper proposes an interesting and promising method to learn an encoder-decoder with a novel two-player minimax game, the paper is not yet well enough written and lacks some experimental results that allow for comparison with state of the art methods.

---

> ### Author Response · Authors · 2021-11-10
> **Reply to reviewer hk2J**
>
> Thank you very much for recognizing the novelty of our formulation. We believe you may have some misunderstandings about our work and the ``weaknesses” you pointed out should not justify rejection of our paper. We sincerely hope our explanation below will help change your opinion.
>
> 1. Self-contained: Due to the space limit, the introduction to MCR2 is brief. Realizing it is not necessarily familiar to many readers, we will provide more explanation in the revision. The rate reduction (between two subspace-Gaussians)  measures the volume between them in terms of the number of epsilon-balls packed in between. It is explained in the MCR2 work and references therein and we will provide a clear reference there. Yes, both Z and \hat{Z} are in R^{dxn} representing features of X and \hat{X} respectively. Yes, both (6) and (4) are rate reductions except that (6) is a special (simplified) case for only two sets with equal sample size. We have clarified in the revision.
>
> 2. Experiments. The experiments in Table 1 are only meant to make a fair comparison with the most basic GAN and VAE formulations, with the same experimental conditions (the same data and choice of networks, etc.) We only want to show that as a new formulation, our formulation is as sound as GAN and VAE. We provide extensive quantitative comparison with more recent representative GANs and VAEs  in Table 2 and Table 13, using similar networks and training. The learned auto-encoding is not perfect since we do not impose any direct loss between x and \hat{x}, especially for a diverse dataset like ImageNet. Nevertheless on MNIST, our method is clearly better than others. Also, at the time of the submission, our results on ImageNet were still partial (did not finish enough training iterations). The performance (in terms IS and FID) has improved with more training iterations. BTW, by Figure 14, you mean Figure 13 in the appendix? The images in each row are supposed to be similar as they are sampled along the same principal component, and the diversity is captured in different rows as different principal components for each class.
>
> 3. Note that in this work, our main goal is to demonstrate the potential of this novel formulation for learning generative models, instead of competing for state-of-the-art performance. Nevertheless, we are comparing our formulation with GANs for their known strength in visual quality and with VAEs for their strength in discriminative properties.  One of the strengths of our method is to naturally handle multiple classes in learning generative models. Note that all generative methods are not specifically engineered for classification tasks. For instance, we are not aware of any VAE methods that report classification performance on CIFAR10 and yet ours is above 80%. We trained our model on ImageNet by treating all classes as one and then refined it onto only 10 classes (due to the limit of computational resources). The computational complexity of our method scales linearly in the number of classes. Indeed, if one wishes the learned features for different classes to be disentangled (independent), the dimension of the feature space has to grow linearly in the number of classes. Given the simplicity in our formulation and current implementation, we are confident that with more computational resources and engineering effort, this novel formulation has great potential to push the envelope of representation learning for both generative and discriminative purposes in the future.
>
> Regarding minor issue. Thanks for your suggestions and we will correct these points in our revision. In particular, our comments about the relationship between GAN and WGAN are indeed somewhat misleading. What we meant to say is: the W1-distance is introduced as a better surrogate for the JS-divergence of GAN as the JS-divergence becomes ill-conditioned to optimize when the supports of two distributions are low-dimensional and do not overlap. We have corrected this in our revision. We put (\hat{X}, X) together in the diagram to indicate they are typically compared as a pair at that stage, not as output of g. We can change that notation in the revision to avoid confusion.

---

### Official Review · Reviewer_ykKP · 2021-11-03

**Correctness:** 4
**Technical Novelty And Significance:** 3
**Empirical Novelty And Significance:** Not applicable
**Recommendation:** 6
**Confidence:** 3

**Main Review:**

Strengths:

(1) The proposed adversarial objective function is novel and interesting.

(2) The experimental results are solid.

Weakness:

(1) No background is given on rate reduction and MCR2. As a result, the paper is not self-contained, and is not easy to understand.

(2) The objective function is less simple and clean than VAE and GAN loss functions, which are closely related to log-likelihood of a generative model or a discriminative model. The objective function in this paper is quite complex.

(3) No theoretical analysis of the proposed method is given.

(4) When generating a new x, is z sampled from N(0,I)? The probabilisitic generative model is not very explicit in this paper. A probabilistic generative/decoder model is a natural representation of the observed data, with encoder being interpreted as approximated posterior inference. The auto-encoding point of view appears rather limited in comparison.


**Summary Of The Paper:**

This paper proposes to learn auto-encoder using an adversarial objective based on rate reduction. The experiments show that the proposed method achieves competitive performance on reconstruction, generation, and discrimination.

**Summary Of The Review:**

The paper is an extension of MCR. It appears a bit complex.

---

> ### Author Response · Authors · 2021-11-10
> **Reply to reviewer ykKP**
>
> Thank you very much for recognizing the novelty of our formulation. Below we clarify the "weaknesses" you pointed out in your comments.
>
> (1). Due to the limited space of the conference paper, our introduction to MCR2 is terse as it is prior work by others. We have added more explanation about the MCR2 objective in the revision.
>
> (2) Although conceptually the objective functions of VAE and GAN seem simple, in practice there are many caveats to compute or optimize them directly. In VAE the KL divergence between distributions typically has to be approximated by variational bounds; and in GAN the JS-divergence is impossible to optimize directly when the support of the distribution is degenerate and non-overlapping, hence the introduction of W1-distance or many other variants. Note that, even between two Gaussians, there is no closed-form formula for the JS-divergence nor the W1-distance. They have to be approximated in practice. As a result, the actual objective functions of almost all modern VAEs and GANs include multiple heuristics or regularizing terms.
>
> In contrast, in our objective function, all terms are conceptually simple and uniform: they are all rate reductions between subspace-like Gaussians; furthermore, they are all explicitly given in closed-form, without any approximation needed. There are no other heuristic or regularizing terms in our objective function at all.
>
> (3) As we have discussed in Section 2.4, our formulation is based on ample prior theoretical work. The goal of this paper is to provide empirical verification of this new framework instead of claiming any new theoretical contributions. We believe the simplicity and novelty of our formulation and the compelling empirical results are very valuable to the community.
>
> (4) In fact, one of the main benefits of our formulation is precisely to obtain an *explicit and disentangled* distribution for the feature. The learned feature z is a mixture of independent subspace-Gaussians: one subspace per class. More details about how z is explicitly modeled and sampled in the feature space can be found in Appendix A.2 around equation (13). Furthermore, one may even view that such a model naturally disentangles different classes as independent subspaces and disentangles different visual attributes of each class as principal components within each subspace, see Figures 11, 13, and 14 in the Appendix. In contrast, the posterior distributions of the feature p(z|x) obtained from GANs or VAEs usually do not have such an explicit interpretation. The distribution of z (both its support and density) inside the feature space remains “hidden” and even “entangled”.

---

### Official Review · Reviewer_beUC · 2021-11-04

**Correctness:** 3
**Technical Novelty And Significance:** 2
**Empirical Novelty And Significance:** 2
**Recommendation:** 5
**Confidence:** 4

**Main Review:**

The idea of viewing the encoder and the decoder in a close-loop auto-eoncoder as generator and discriminator is quite interesting.

One of the critical parts of the proposed method is viewing the encoder as a discriminator and measuring the distance between X and \hat{X} as the maximization of the rate reduction.  The authors explain that the maximization can avoid that g\circ f is not an auto-encoding map. However, it is still not super clear that why such maximization is reasonable to make g(f(x)) = x. In other words, why not use eqn (7) together with conventional reconstruction loss. This seems like also provide desired maps between multi-class data and LDR. It is a bit hard for me to see the advantages of using maximization there.

Some minor issues:
In eqns (2), Z maps to \hat{X}, it is a bit strange to write g maps Z to (\hat(X),X). Similar issue for eqn (5).

In Section 1.2, X has been amused for different sets.



**Summary Of The Paper:**

The manuscript considers learning a closed-loop auto-encoder between multi-class multi-dimension data and LDR. It also provides various experiments to verify the proposed method.

**Summary Of The Review:**

The explanation of the maximization part or two-player game part is not super clear to me.

---

> ### Author Response · Authors · 2021-11-10
> **Reply to reviewer beUC**
>
> Thank you very much for recognizing the novelty of our formulation, as do the other two reviewers. Below we provide specific answers to your concerns:
>
> We consider maximizing the rate reduction only in the feature space, using eqn (8), as one of the most distinctive features of this new formulation. As we discussed earlier in the Introduction above eqn (3), conventional reconstruction losses in the image (X) domain, say those used in autoencoding, do not give a good similarity in visual quality, which is the reason why GAN adopts a discriminator or why many recent VAE based methods use an additional discriminator to improve visual quality.
>
> Here, by maximizing the rate reduction of the features, we allow the encoder to play the same role as the discriminator in GAN hence we need *only two* networks, instead of more networks as in most recent VAE+GAN architectures.  Intuitively, as stated in our paper, the maximization promotes the encoder f to be injective by discerning any fine differences (in dimension, support) between the original data X and the decoded \hat{X} through their mapped features f(X) and f(\hat{X}). Also, the rate reduction is a more refined (detail-preserving, discriminative) measure than the cross-entropy typically used in GANs, which has been established in the work of Yu et. al. 2020. We have done additional ablation studies with losses such as the cross-entropy, which shows the superiority of the rate reduction. We can include that in the revised version.
>
> Regarding the diagrams in (2) and (5), we use (\hat{X}, X) only to illustrate that they are typically to be compared as a pair in those settings. We have changed the notation in the revision as that seems to cause some confusion.
>
> The X in Section 1.1 and 1. 2 both represent the sample dataset, only that in Section 1.2 X consists of k subsets.

---

### Author Response · Authors · 2021-11-10
**Comments to all reviewers and AC**

We thank all three reviewers for unanimously recognizing the novelty of our formulation. This new approach is based on a relatively recent work of MCR2 by others that may not be so familiar to the readers, we will provide more explanation in the revised version to be more self-contained. Some of you seem to have misunderstood or overlooked the contribution and significance of our work, such as 1. simplicity in our objective function, 2. explicit representation of the learned features, and 3. solid experimental comparison with recent methods. We hope with our detailed clarification of your concerns and careful revision, you will find our work well-qualified for this conference.

About the revised version:
We have modified our paper based on the concerns of the reviewers and have uploaded a revised version. All revised places are marked in red color. Besides fixing some of the minor changes suggested by the reviewers, we have made three main updates:

1. For reviewers ykKP and hk2J, we have provided more explanation about MCR2 to make things more self-contained.
2. For reviewer hk2J, we have updated experimental results on STL-10 and ImageNet in Table 2 and Table 13 for comparison with recent state of the art methods. More visualizations are provided in Appendix A.5 for STL-10 and Appendix A.7 for ImageNet.
3. For reviewer beUC, we have conducted a new ablation study about using the more conventional loss, cross-entropy, with the closed-loop setting. The results are added as Appendix A.8 and they suggest the role of rate reduction is quite crucial.

We hope the revision has addressed all your concerns. Please let us know if you have any further suggestions or questions.

---

### Decision · Program_Chairs · 2022-01-20

**Decision:**

Reject

**Comment:**

The authors describe an approach to modeling data via an implicit representation that lives on a union of linear subspaces.  While the reviewers consider the authors' approach as novel and having potential, they (and myself) consider that the exposition and notation could be improved, and that the paper as it is is hard to understand and contextualize.